# MIGA: Mutual Information-Guided Attack on Denoising Models for Semantic Manipulation

## Abstract

Deep learning-based denoising models have been widely employed in vision tasks, functioning as filters to eliminate noise while retaining crucial semantic information. Additionally, they play a vital role in defending against adversarial perturbations that threaten downstream tasks. However, these models can be intrinsically susceptible to adversarial attacks due to their dependence on specific noise assumptions. Existing attacks on denoising models mainly aim at deteriorating visual clarity while neglecting semantic manipulation, rendering them either easily detectable or limited in effectiveness. In this paper, we propose Mutual Information-Guided Attack (MIGA), the first method designed to directly attack deep denoising models by strategically disrupting their ability to preserve semantic content via adversarial perturbations. By minimizing the mutual information between the original and denoised images—a measure of semantic similarity—MIGA forces the denoiser to produce perceptually clean yet semantically altered outputs. While these images appear visually plausible, they encode systematically distorted semantics, revealing a fundamental vulnerability in denoising models. These distortions persist in denoised outputs and can be quantitatively assessed through downstream task performance. We propose new evaluation metrics and systematically assess MIGA on four denoising models across five datasets, demonstrating its consistent effectiveness in disrupting semantic fidelity. Our findings suggest that denoising models are not always robust and can introduce security risks in real-world applications.

## 1 Introduction

Image denoising is a fundamental task in computer vision, aiming to recover visually clean images from noisy observations Rudin et al. (1992); Buades et al. (2005). Beyond restoring image clarity, modern denoising models also strive to preserve *critical semantic information* vital for downstream tasks such as autonomous driving Guoqiang et al. (2023); Kloukiniotis et al. (2022), medical diagnosis Kaur et al. (2018); Jifara et al. (2019), and remote sensing Rasti et al. (2021); Singh & Shankar (2021). Moreover, denoising has been adopted as a defense mechanism to remove adversarial noise before it reaches the final model Liao et al. (2018); Bakhti et al. (2019). Recently, the advent of deep learning has led to significant advancements in denoising models, setting new benchmarks across various datasets. Zhang et al. (2021; 2019); Li et al. (2023); Liang et al. (2021); Zamir et al. (2022).

However, despite these advances, deep learning-based denoising models remain inherently vulnerable and can become targets of adversarial attacks, compromising two key aspects of their outputs: clarity and semantic fidelity Ning et al. (2023). Specifically, clarity refers to low-level perceptual quality, which is typically assessed using image quality metrics such as PSNR and SSIM Wang & Bovik (2002); Goodfellow et al. (2014), while semantic fidelity refers to the preservation of high-level content, which needs to be evaluated based on downstream task performance. Traditional adversarial attacks, which were originally designed for classifiers and general vision models rather than denoising pipelines, primarily aim to degrade clarity metrics while largely ignoring targeted manipulation of semantics Szegedy (2013). Additionally, these attacks frequently introduce visible artifacts, making them easily detectable. These limitations reduce their effectiveness in real-world scenarios.

As illustrated in Fig. 1, a noisy "No Straight" sign $A$ may contain both natural noise and adversarial perturbations. In the typical case (a), the denoiser restores it to a clean and correct form, allowing the Automated Driving Model (ADM) to make the appropriate "Turning Decision". In contrast,

traditional adversarial attacks, as shown in case (b), mainly degrade image clarity while preserving semantic content. As a result, the ADM can still recognize the intended meaning and respond correctly.

Moreover, if the degradation becomes too severe, the system may trigger an "Unrecognized Image Warning", thereby preventing an incorrect decision.

Moreover, if the degradation becomes too severe, the system may trigger an "Unrecognized Image Warning", thereby preventing an incorrect decision.

To enhance the effectiveness of adversarial attacks on denoising models in real-world scenarios, we identify two key challenges. **(1)** The attack should avoid introducing noticeable artifacts on denoised images, preserving a high-quality visual appearance that is less likely to be flagged or rejected. **(2)** More importantly, it must subtly alter the semantic content rather than merely degrading image clarity, leading to

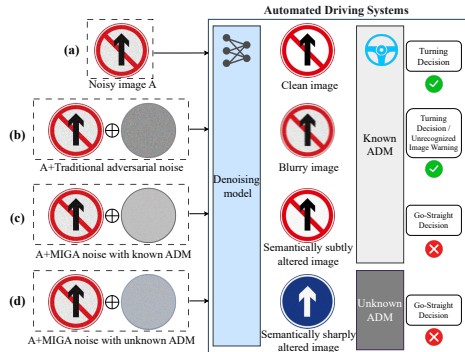

Figure 1: Conceptual overview of our threat model for adversarial attacks on automated driving systems with a denoising module.

semantic discrepancies that affect downstream model predictions. To address these challenges, we propose the *Mutual Information-Guided Attack* (MIGA), a novel adversarial framework that introduces imperceptible perturbations to manipulate semantic features during the denoising process by minimizing the mutual information (MI) Shannon (1948) between the original and denoised images. MI quantifies the amount of shared semantic content between two variables, and in the context of denoising, it specifically reflects the extent to which semantic information is retained after noise removal Zhu & Wu (2005). By introducing carefully crafted perturbations, MIGA minimizes MI through a mutual information loss jointly optimized with a perturbation constraint loss and a reconstruction loss, as illustrated in Fig. 2. This induces the denoiser to generate visually clean images with systematically altered semantics, ultimately undermining its reliability and propagating corrupted semantics to downstream tasks.

We further distinguish between known and unknown downstream tasks. A task is known if the attacker has prior knowledge of its objective and model architecture, enabling tailored perturbations. Otherwise, it is unknown, making targeted semantic shifts challenging. To address this, MIGA adapts its strategy accordingly. In the known setting, it analyzes the impact of semantic shifts on task performance to craft perturbations that make the denoiser produce a visually clean yet subtly altered image. In the unknown setting, the lack of task-specific guidance prevents direct optimization. To overcome this, MIGA introduces a reference image to guide semantic alignment, enabling controlled semantic modifications without explicit knowledge of the downstream model. As shown in Fig. 1(c), in the known scenario, MIGA manipulates the denoiser to produce a denoised "No Straight" sign that appears visually clean but has subtly altered semantics, leading the ADM to make an incorrect "Go Straight Decision". In the unknown scenario (Fig. 1(d)), MIGA leverages a "Go Straight" reference image to guide semantic shifts, resulting in similarly misleading downstream decisions.

In summary, our main contributions include:

- **First semantic adversarial attack on denoising models.** We propose MIGA to specifically disrupt *semantic fidelity* while preserving high visual quality, revealing a previously overlooked vulnerability in deep denoising models.

- **Task-relevant MI formulation.** We introduce MI-based loss terms to systematically reduce the task-relevant semantic overlap between the original and denoised images, effective in both known and unknown task scenarios.

- **Extensive experiments & tailored metrics.** We validate MIGA on four denoising models and five datasets, designing new evaluation metrics to measure semantic manipulation. Results confirm MIGA's ability to produce perceptually clean yet semantically disruptive outputs that mislead real-world downstream tasks.

## 2 RELATED WORK

**Image denoising.** Image denoising is essential for applications such as autonomous driving Guoqiang et al. (2023); Kloukiniotis et al. (2022), medical diagnosis Kaur et al. (2018); Jifara et al. (2019), and remote sensing Rasti et al. (2021); Singh & Shankar (2021), where clear visual data is crucial. Moreover, denoising has been widely adopted as a defense mechanism to remove adversarial noise before it reaches the final model Liao et al. (2018); Bakhti et al. (2019). Traditional denoising approaches such as total variation Rudin et al. (1992), Non-Local Means Buades et al. (2005), and sparse coding Mairal et al. (2009) established early foundations. With the advent of deep learning, performance has significantly improved using CNN- Zhang et al. (2017), non-local Zhang et al. (2019), and Transformer-based Liang et al. (2021); Zamir et al. (2022) architectures. Recent studies emphasize that preserving semantic information during denoising is critical for downstream task performance Buchholz et al. (2020). However, despite these advances, deep learning-based denoising models remain susceptible to adversarial attacks, which can compromise their ability to maintain both image clarity and semantic fidelity Ning et al. (2023).

**Mutual information.** MI quantifies the shared information between two variables Shannon (1948); Hjelm et al. (2019); Tu et al. (2021) and is widely utilized in tasks such as image registration Maes et al. (1997), feature selection Peng et al. (2005), and representation learning Hjelm et al. (2019). In the context of image denoising, MI serves as a measure of dependency between noisy and clean images Zhu & Wu (2005), guiding models to retain essential semantic information while removing noise. For example, denoising autoencoders leverage MI to learn robust representations that preserve semantic content Vincent et al. (2010). However, prior work has primarily focused on maximizing MI for semantic preservation, while its potential for adversarial manipulation in denoising remains largely unexplored.

**Adversarial attacks on denoising models.** Adversarial attacks add small, often imperceptible perturbations to inputs to induce errors Goodfellow et al. (2014). Beyond classifiers, denoising models are likewise vulnerable Ryou et al. (2024); Ning et al. (2023). Most prior attacks on denoisers target entropy reduction and thus degrade image clarity Agnihotri et al. (2023); Yan et al. (2022), often leaving visible artifacts and overlooking semantic manipulation. Pasadena Cheng et al. (2021) addresses semantics but relies on customized architectures, limiting generality. In contrast, we present a task-agnostic framework that manipulates mutual information to yield imperceptible yet semantically altered adversarial examples, without modifying architectures and with broad applicability to standard denoisers and downstream tasks.

## 3 PROBLEM SETUP

Consider a clean image $x \in \mathbb{R}^n$. This image is corrupted by noise $\eta$ and yields a noisy image $x_\mathrm{N} = x + \eta$. The clean image $x$ is unknown, while the noisy image $x_\mathrm{N}$ is given. There is a denoising model $D$ that aims to recover the clean image from the noisy image $x_\mathrm{N}$ such that $D(x_\mathrm{N}) \approx x$. After the image has been denoised, there is a downstream task (e.g., classification, regression, human perception) that uses this denoised image to determine certain outputs (e.g., label, regression outcome). Let $F$ denote the downstream task. Let $C$ denote the ground-truth output of downstream task $F$ by inputting the clean image $x$, i.e., $F(x) = C$.

In this work, we aim to find an adversarial perturbation $\delta$ added to the noisy image $x_\mathrm{N}$, resulting in a modified image $x_\mathrm{N} + \delta$, such that the denoised output $D(x_\mathrm{N} + \delta)$ retains high clarity but alters semantic content to mislead the downstream task model $F$. Formally, our goal is to find an adversarial perturbation $\delta$ that satisfies the following objectives:

1. **Imperceptible Perturbation**: $\delta$ should be small and imperceptible to human observers, ensuring that $x_\mathrm{N} + \delta$ is visually indistinguishable from $x_\mathrm{N}$.

2. **Image Clarity Preservation**: The denoised image $D(x_\mathrm{N} + \delta)$ should maintain high visual quality, appearing as a clean image without noticeable artifacts.

3. **Semantic Alteration**: The semantic information in $D(x_\mathrm{N} + \delta)$ should be altered to mislead the downstream task model $F$, i.e., $F(D(x_\mathrm{N} + \delta))$ differs from $C$.

In this work, we consider both cases where the downstream task model $F$ is known and unknown.

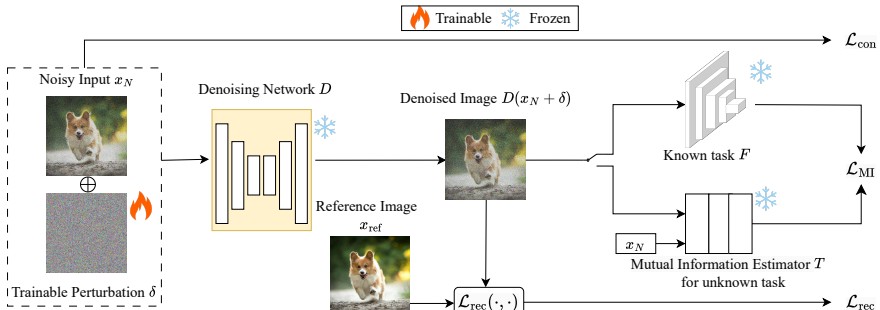

Figure 2: Architecture overview of MIGA. $\mathcal{L}_{\text{con}}$ enforces the imperceptible perturbation constraint; $\mathcal{L}_{\text{rec}}$ ensures image clarity using a clean reference image $x_{\text{ref}}$; $\mathcal{L}_{\text{MI}}$ characterizes the task-relevant mutual information. When the downstream task is known, the mutual information is estimated using $F$; when the downstream task is unknown, it is estimated using a mutual information estimator $T$.

## 4 METHODOLOGY

In this section, we first propose a novel task-relevant mutual information that characterizes the degree of matching between the clean image $x$ and the denoised image $D(x_{\text{N}} + \delta)$, which will be used for quantifying the semantic alteration. Unlike the traditional mutual information metric, our proposed metric can evaluate how much downstream task-relevant information is preserved in image denoising. Then, we propose the Mutual Information Guided Attack (MIGA) to attack denoising models by designing three carefully-crafted loss functions and the optimization process.

### 4.1 TASK-RELEVANT MUTUAL INFORMATION

Traditional mutual information $I(x; D(x_{\text{N}} + \delta))$ measures the shared information between the original image $x$ and the denoised output $D(x_{\text{N}} + \delta)$, considering the adversarial perturbation. This metric, however, encompasses both downstream task-relevant and irrelevant details. In this work, to emphasize on the semantic features that are important for the downstream task, we introduce the concept of task-relevant mutual information. This metric quantifies the shared information between the original image $x$ and the denoised image $D(x_{\text{N}} + \delta)$, conditioned on the ground-truth output $C$ of the downstream task. For example, in an image of a landscape containing a person, the task-relevant mutual information for location recognition would primarily focus on the background features, whereas for face recognition, it would emphasize facial details. Specifically, task-relevant mutual information is defined as follows:

$$I(x; D(x_{\text{N}} + \delta) \mid C) = H(C \mid x) - H(C \mid D(x_{\text{N}} + \delta)), \tag{1}$$

where $H(C \mid x)$ and $H(C \mid D(x_{\text{N}} + \delta))$ represent the conditional entropies of the ground-truth $C$ given the original and denoised images, respectively.

This formulation helps us assess how much task-relevant information is preserved in the denoised image. Thus, we can achieve semantic alteration by minimizing task-relevant mutual information $I(x; D(x_{\text{N}} + \delta) \mid C)$. However, directly computing $I(x; D(x_{\text{N}} + \delta) \mid C)$ is impossible because the clean image $x$ is unknown and $H(C \mid D(x_{\text{N}} + \delta))$ involves high-dimensional probability distribution estimation. In the following, we present methods to measure or approximate the task-relevant mutual information for known and unknown downstream task scenarios, where these measures will be used in loss function design.

**Known downstream task.** When the downstream task model $F$ is known, we propose to use the cross-entropy loss $\mathcal{L}_{\text{CE}}(F(D(x_{\text{N}} + \delta)), C)$ to characterize the task-relevant mutual information. We prove that maximizing such a cross-entropy loss is equivalent to minimizing the task-relevant mutual information $I(x; D(x_{\text{N}} + \delta) \mid C)$, with a detailed proof provided in Appendix A.

**Unknown downstream task.** When the downstream task $F$ is unknown, we propose a semantic alteration strategy to estimate the task-relevant mutual information without requiring explicit task-specific knowledge. Specifically, we first construct a dataset of paired images $(x, x_{\text{alt}})$, where $x_{\text{alt}}$ serves as a reference image with altered semantics compared to the original image $x$. The alteration process involves modifying key semantic regions of $x$, such as objects, scenes, or textual content, which alters the image's semantics. In practice, the generation of such reference images $x_{\text{alt}}$ can be

encapsulated as an automatic pre-processing module in our attack pipeline. Given an input image $x$ and a desired semantic target, this module either applies simple edits to $x$ (e.g., masking or replacing selected regions) or invokes modern diffusion-based semantic editing tools that modify specific objects or text while largely preserving the rest of the scene Couairon et al. (2022); Cao et al. (2023); Yang et al. (2023). From the perspective of an attacker or a privacy-conscious user, it is therefore sufficient to specify the target semantics (e.g., turning a "stop" sign into a "go-straight" sign or sanitizing personal information in uploaded photos), while the algorithm automatically produces suitable $(x, x_{\text{alt}})$ pairs in a (semi-)automatic manner. The resulting paired dataset is then used to pre-train the MINE network Belghazi et al. (2018). The network learns to estimate the task-relevant mutual information by modeling the joint distribution between the original image $x$ and its altered version $x_{\text{alt}}$, enabling it to capture relevant semantic information without the need for explicit task labels.

Specifically, we train a neural network estimator $T$ to distinguish between positive pairs $(x, D(x_{\text{N}}))$ and negative pairs $(x, x_{\text{alt}})$. We use a contrastive loss to encourage the model to assign higher similarity scores to positive pairs and lower scores to negative pairs, focusing on the model on the image's semantically relevant content. After training, $T(x, D(x_{\text{N}}))$ measures the similarity between the original image $x$ and its denoised version $D(x_{\text{N}})$, preserving the core structure and semantics of the image. By optimizing the adversarial perturbation $\delta$ to minimize $T(x, D(x_{\text{N}} + \delta))$, task-relevant mutual information can be approximately minimized.

### 4.2 LOSS FUNCTIONS

According to the objectives in Sec. 3, we propose to consider three losses: $\mathcal{L}_{\text{con}}(x_{\text{N}}, x_{\text{N}} + \delta)$ enforces the imperceptible perturbation constraint; $\mathcal{L}_{\text{rec}}(x_{\text{ref}}, D(x_{\text{N}} + \delta))$ ensures image clarity using a clean reference image $x_{\text{ref}}$; $\mathcal{L}_{\text{MI}}$ characterizes the task-relevant mutual information. In other words, the objective can be transformed into optimizing the adversarial perturbation $\delta$ such that

$$\delta^* = \arg\min_{\delta} \left[ \lambda_{\text{con}} \mathcal{L}_{\text{con}}(x_{\text{N}}, x_{\text{N}} + \delta) + \lambda_{\text{rec}} \mathcal{L}_{\text{rec}}(x_{\text{ref}}, D(x_{\text{N}} + \delta)) + \lambda_{\text{MI}} \mathcal{L}_{\text{MI}} \right]. \quad (2)$$

We omit the explicit constraint $\|\delta\|_{\infty} \leq \epsilon$, because the perturbation loss $\mathcal{L}_{\text{con}}(x_{\text{N}}, x_{\text{N}} + \delta)$ has already enforced the imperceptibility of $\delta$.

To implement this objective, we design three carefully-crafted loss functions to guide the optimization process.

**Perturbation constraint loss.** To ensure imperceptibility perturbation, we define the following loss function:

$$\mathcal{L}_{\text{con}}(x_{\text{N}}, x_{\text{N}} + \delta) = \alpha \|x_{\text{N}} - (x_{\text{N}} + \delta)\|_2^2 + (1 - \alpha)\mathcal{L}_{\text{perc}}(x_{\text{N}}, x_{\text{N}} + \delta), \quad (3)$$

where $\mathcal{L}_{\text{perc}}(x_{\text{N}}, x_{\text{N}} + \delta)$ is the perceptual loss, computed using feature maps from a pre-trained VGG16 network Simonyan & Zisserman (2014), and $\alpha \in [0, 1]$ controls the trade-off between pixel-level and perceptual similarities.

**Reconstruction loss.** The reconstruction loss ensures that $D(x_{\text{N}} + \delta)$ remains visually similar to a reference image $x_{\text{ref}}$. Its expression is given as follows:

$$\mathcal{L}_{\text{rec}}(x_{\text{ref}}, D(x_{\text{N}} + \delta)) = \beta \|x_{\text{ref}} - D(x_{\text{N}} + \delta)\|_2^2 + (1 - \beta)\mathcal{L}_{\text{perc}}(x_{\text{ref}}, D(x_{\text{N}} + \delta)), \quad (4)$$

where $\beta \in [0, 1]$ balances the pixel-wise and perceptual similarities. The reference image $x_{\text{ref}}$ is selected based on the downstream task: for a known downstream task, $x_{\text{ref}} = D(x_{\text{N}}) \approx x$; for an unknown downstream task, $x_{\text{ref}} = x_{\text{alt}}$.

**Mutual information loss.** The mutual information loss $\mathcal{L}_{\text{MI}}$ is designed to quantify the task-relevant mutual information, as described in Sec. 4.1.

---

**Algorithm 1** Training Process of MIGA

---

1: **Input:** Noisy image $x_\text{N} = x + \eta$, clean image $x$, denoising model $D$, task-specific reference $x_\text{ref}$, perturbation initialization $\delta = 0$, learning rate $\gamma > 0$
2: **Output:** Adversarial perturbation $\delta$
3: **while** not converged **do**
4:     Create adversarial image $x_\text{MIGA} = x_\text{N} + \delta$;
5:     Obtain denoised output $y = D(x_\text{MIGA})$;
6:     Compute $\mathcal{L}_\text{con}(x_\text{N}, x_\text{adv})$ using equation 3;
7:     Compute $\mathcal{L}_\text{rec}(x_\text{ref}, y)$ using equation 4;
8:     Compute $\mathcal{L}_\text{MI}$ using equation 5 for known task and using equation 6 for unknown task;
9:     $\mathcal{L}_\text{total} = \mathcal{L}_\text{con}(x_\text{N}, x_\text{adv}) + \mathcal{L}_\text{rec}(x_\text{ref}, y) + \mathcal{L}_\text{MI}$;
10:    Update perturbation $\delta \leftarrow \delta - \gamma \nabla_\delta \mathcal{L}_\text{total}$;
11: **end while**

---

For a *Known Downstream Task*, the mutual information loss is formulated as:

$$\mathcal{L}_\text{MI} = -\mathcal{L}_\text{CE}(F(D(x_\text{N} + \delta)), C), \tag{5}$$

where $\mathcal{L}_\text{CE}$ is the cross-entropy loss.

For an *Unknown Downstream Task*, the mutual information loss is given by

$$\mathcal{L}_\text{MI} = T(x, D(x_\text{N} + \delta)), \tag{6}$$

where $T(x, D(x_\text{N} + \delta))$ outputs a similarity score between $x$ and $D(x_\text{N} + \delta)$. Minimizing this score reduces the task-relevant mutual information.

### 4.3 TRAINING PROCEDURE

We illustrate our overall framework in Fig. 2, with the detailed process shown in Algorithm 1. To optimize the perturbation $\delta$, we employ a gradient-based approach. We start with a noisy image $x_\text{N} = x + \eta$, where $x$ is the clean image and $\eta$ is the added noise. The perturbation $\delta$ starts at zero and is iteratively updated until convergence.

In each iteration, the adversarial image $x_\text{MIGA} = x_\text{N} + \delta$ is created. The denoised output $y = D(x_\text{MIGA})$ is obtained using the given pre-trained denoising model $D$. The total loss is a weighted sum of the perturbation constraint loss, reconstruction loss, and mutual information loss, computed based on the task. For known tasks, cross-entropy loss is used. For unknown tasks, mutual information is estimated. The perturbation $\delta$ is updated through gradient descent with respect to the total loss, with $\gamma$ being the step size.

## 5 EXPERIMENTS

In this section, we evaluate MIGA on denoising models under both known and unknown downstream tasks. We begin by describing the datasets, metrics, models, and baselines. We then report results together with an ablation study that quantifies the contribution of each component. Experimental settings and implementation details are provided in Appendix C, and additional analyses—including hyperparameter choices, transferability, and extra baselines—are presented in Appendix D.

### 5.1 EXPERIMENTS SETTING

**Datasets.** To evaluate our method, we define specific dataset requirements based on the nature of downstream tasks, which performance serve as indicators for semantic modification. For **known downstream tasks**, a well-defined task specification and the availability of high-performing pre-trained models are essential. We use the ImageNet-10 dataset Russakovsky et al. (2015), which consists of a 10-class classification task and is supported by several state-of-the-art pre-trained models. For **unknown downstream tasks**, a clean reference image with modified semantics is required. We employ the Tampered-IC13 dataset Wang et al. (2022), which includes real-world text alterations in natural scenes, and the MAGICBRUSH dataset Zhang et al. (2024), originally designed for image

editing tasks, containing both the original images, editing instructions, and the modified results. To further enhance the generalizability of our evaluation, we incorporate two synthetic datasets. The first is a stylized dataset created using the neural style transfer method Gatys (2015), which includes both the original images and their corresponding stylized reference images. Additionally, we create a synthetic text alteration dataset by selecting a subset of images from the SSDI dataset Abdelhamed et al. (2018) as backgrounds, and generating pairs of small but semantically distinct text fragments, placed at the same location in each background. Overall, these public datasets provide paired samples $(x, x_{\text{ref}})$ with well-defined semantic edits (e.g., text alteration, content replacement, style transfer), which align with our goal of semantic modification without requiring access to the downstream task model. We use them purely for reproducibility and controlled evaluation, as convenient, standardized, and publicly verifiable instantiations of the desired target semantics. In all experiments, Gaussian noise was added to the images in different datasets to generate noisy versions. Specifically, a Gaussian noise with a variance of 50 was applied to the ImageNet-10 dataset, while Gaussian noise with a variance of 25 was added to other datasets. In the calculation of $\mathcal{L}_{\text{rec}}$ during the experiment, the reference image $x_{\text{ref}}$ used was the clean image, without any added noise.

**Evaluation metrics**. To comprehensively evaluate our algorithm, we define a set of metrics targeting three key objectives: (1) the imperceptibility of perturbations, (2) the clarity of denoised images, and (3) the semantic integrity of denoised images. For **imperceptibility**, we compute the LPIPS score Zhang et al. (2018) between the noisy image $x_{\text{N}}$ and the perturbed image $x_{\text{N}} + \delta$. LPIPS measures the perceptual similarity between images, with lower values indicating higher visual similarity. We use LPIPS$_{\text{con}}$ to denote this perceptual similarity constraint. For **clarity**, we measure the similarity between the denoised image and the clean reference using multiple indicators: PSNR, SSIM Wang & Bovik (2002), LPIPS Zhang et al. (2018), and image entropy Prashanth et al. (2023). Higher PSNR and SSIM values, combined with lower LPIPS and entropy scores, reflect better denoising quality and improved image clarity. To evaluate **semantic modification**, we employ different approaches based on whether the downstream task is known.

For **known downstream tasks**, performance degradation signals semantic changes. For example, in the classification task, a drop in accuracy suggests potential semantic alterations. For **unknown downstream tasks**, we assess semantic fidelity by measuring core performance changes in the reference image. In the **text tampering** dataset, OCR Singh et al. (2012) is used to extract text from denoised images, and ROUGE-L Lin (2004) scores between the extracted and reference text quantify semantic shifts. In **image editing** datasets, CLIP similarity Radford et al. (2021) between the generated image and editing instructions gauges adherence to the intended modification. For **style alteration** tasks, we use Gram matrix loss Gatys (2015) to compare the denoised image with the stylized reference image, where a lower loss indicates better style alignment.

**Models and baselines.** To extensively evaluate our experiments, we select several different denoising network architectures, including Xformer Zhang et al. (2023), Restormer Zamir et al. (2022), PromptIR Potlapalli et al. (2024), and adversarially robust AFM Ryou et al. (2024), all of which come with open-source pre-trained models. For downstream tasks involving classification, we use a pre-trained ResNet50 He et al. (2016) as the classifier. In scenarios with unknown downstream tasks, we first train a MINE network as described in Sec. 4.1. Since existing adversarial attacks on denoising models focus on degrading image clarity rather than explicitly manipulating semantics, we adopt the traditional adversarial attack method I-FGSM Kurakin et al. (2018) as a representative baseline. The corresponding adversarial image is $x_{\text{adv}}$ and, after applying the denoising process, we get the denoised image $D(x_{\text{adv}})$.

Table 1: Summary results for Restormer on ImageNet-10.

| Network | Image | PSNR↑ | SSIM↑ | LPIPS↓ | Entropy↓ | Accuracy↓ |
|---|---|---|---|---|---|---|
| | $x$ | $\infty$ | 1.00 | 0.00 | 7.19 | 99.46 |
| | $x_{\text{N}}$ | 17.65 | 0.30 | 0.62 | 7.71 | 83.00 |
| Xformer | $D(x_{\text{N}})$ | 23.63 | 0.65 | 0.41 | 7.04 | 84.73 |
| | $x_{\text{adv}}$ | 17.07 | 0.28 | 0.66 | 7.73 | 78.50 |
| | $D(x_{\text{adv}})$ | 16.04 | 0.34 | 0.66 | 7.20 | 39.23 |
| | $x_{\text{MIGA}}$ | 17.62 | 0.30 | 0.63 | 7.72 | 74.69 |
| | $D(x_{\text{MIGA}})$ | 25.64 | 0.67 | 0.22 | 7.13 | 35.23 |
| Restormer | $D(x_{\text{N}})$ | 22.36 | 0.70 | 0.38 | 6.81 | 89.85 |
| | $x_{\text{adv}}$ | 17.21 | 0.29 | 0.65 | 7.72 | 76.85 |
| | $D(x_{\text{adv}})$ | 15.98 | 0.57 | 0.56 | 6.72 | 64.77 |
| | $x_{\text{MIGA}}$ | 17.60 | 0.30 | 0.63 | 7.72 | 70.62 |
| | $D(x_{\text{MIGA}})$ | 30.77 | 0.78 | 0.26 | 7.06 | 51.54 |
| PromptIR | $D(x_{\text{N}})$ | 22.83 | 0.50 | 0.45 | 7.56 | 85.77 |
| | $x_{\text{adv}}$ | 16.46 | 0.26 | 0.67 | 7.77 | 64.65 |
| | $D(x_{\text{adv}})$ | 11.94 | 0.11 | 0.80 | 7.78 | 34.77 |
| | $x_{\text{MIGA}}$ | 17.61 | 0.30 | 0.63 | 7.72 | 74.46 |
| | $D(x_{\text{MIGA}})$ | 24.20 | 0.59 | 0.37 | 7.60 | 21.31 |
| AFM | $D(x_{\text{N}})$ | 23.01 | 0.71 | 0.42 | 7.01 | 86.43 |
| | $x_{\text{adv}}$ | 17.21 | 0.28 | 0.64 | 7.72 | 79.12 |
| | $D(x_{\text{adv}})$ | 16.09 | 0.58 | 0.61 | 6.92 | 70.27 |
| | $x_{\text{MIGA}}$ | 19.32 | 0.32 | 0.62 | 7.63 | 74.92 |
| | $D(x_{\text{MIGA}})$ | 31.28 | 0.78 | 0.21 | 7.03 | 42.99 |

Table 2: Perturbation constraint for ImageNet-10.

| Metric | Model | $ImageNet$-10$_{\text{adv}}$ | $ImageNet$-10$_{\text{MIGA}}$ |
|---|---|---|---|
| LPIPS$_{\text{con}}$↓ | Restormer | 0.04 | 0.01 |
| | Xformer | 0.04 | 0.01 |
| | PromptIR | 0.07 | 0.01 |
| | AFM | 0.02 | 0.01 |

## 5.2 ATTACK DENOISING MODELS WITH KNOWN TASKS

In this subsection, we evaluate the effectiveness of our proposed method, MIGA, to attack denoising models in scenarios where the downstream task is known, focusing on image classification using the ImageNet-10 dataset. We compare MIGA with I-FGSM, assessing both denoised image clarity and classification accuracy. Tab. 17 summarizes the results, where higher PSNR and SSIM values, along with lower LPIPS and entropy, indicate better image clarity, and a reduction in classification accuracy reflects semantic alteration. The key findings are as follows:

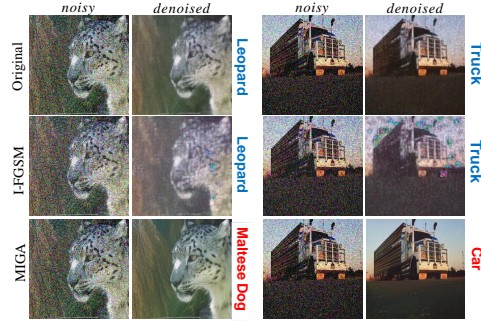

Figure 3: Attack results on ImageNet-10.

**Improvement in downstream task performance by denoising:** Denoising with $D$ improves downstream task performance. For all four models, the accuracy of the noisy image $x_N$ increases significantly after denoising, reaching 84.73%, 89.85%, 85.77%, and 86.43%, respectively.

**Limitations of traditional attacks:** I-FGSM aims to reduce the clarity of denoised images, but this comes at the cost of significant quality degradation. As shown in Fig. 3, it introduces color distortions and artifacts, leading to noticeable image degradation. Despite this, both 'Truck' and 'Leopard' remain correctly classified.

**Effectiveness of MIGA:** MIGA consistently enhances denoised image clarity while notably reducing classification accuracy (see Fig. 3 and Tab. 17). For example, under the Restormer model, MIGA achieves a PSNR of 30.77 and a low LPIPS of 0.26 after denoising $D(x_{MIGA})$, indicating high perceptual image quality. Meanwhile, the classification accuracy drops to 51.54%, demonstrating the effectiveness of our method in altering semantic information during the denoising process without compromising image quality. As illustrated in Fig. 3, MIGA causes the model to generate visually clean yet semantically misleading outputs, leading pre-trained classifiers to misidentify 'Leopard' as 'Maltese Dog' and 'Truck' as 'Car'.

**Imperceptible perturbations:** MIGA introduces minimal perturbations that result in negligible perceptual differences, as shown in Tab. 2. The low LPIPS values (e.g., 0.01 for Restormer) ensure that the attack remains undetected while affecting the downstream task.

These results show that MIGA effectively performs semantic attacks on denoising models in scenarios where the downstream task is known, achieving high-quality denoising while significantly reducing downstream task performance without introducing noticeable artifacts.

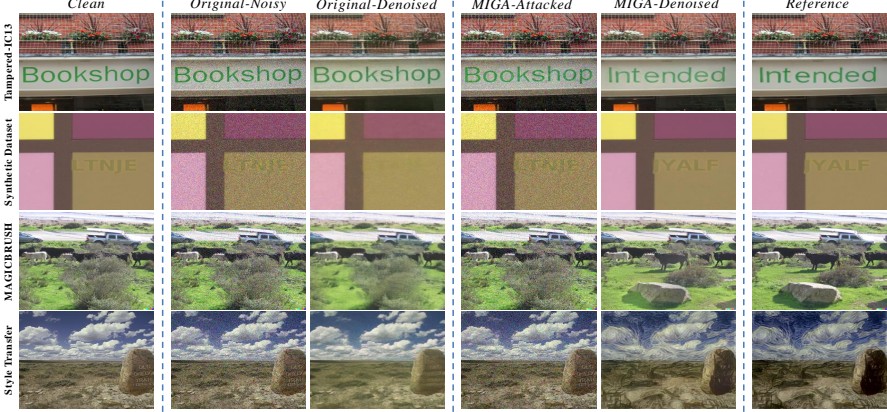

Figure 4: Attack results on unknown tasks. MIGA can induce the image denoising process to shift towards the semantic direction of the reference by adding small perturbations.

Table 3: Denoising Attack for Text Alteration Tasks.

| Network | Image | Tampered-IC13 | | | | | Synthetic Dataset | | | | |
|---------|-------|--------|--------|--------|----------|----------|--------|--------|--------|----------|----------|
| | | PSNR↑ | SSIM↑ | LPIPS↓ | Entropy↓ | ROUGE-L↑ | PSNR↑ | SSIM↑ | LPIPS↓ | Entropy↓ | ROUGE-L↑ |
| Xformer | $x$ | ∞ | 1.00 | 0.00 | 6.60 | 0.57 | ∞ | 1.00 | 0.00 | 5.05 | 0.15 |
| | $x_{\text{N}}$ | 23.98 | 0.47 | 0.48 | 7.32 | – | 24.15 | 0.28 | 0.65 | 6.61 | – |
| | $D(x_{\text{N}})$ | 27.57 | 0.85 | 0.30 | 6.61 | 0.57 | 31.70 | 0.90 | 0.39 | 5.10 | 0.15 |
| | $x_{\text{MIGA}}$ | 22.92 | 0.45 | 0.48 | 7.34 | – | 24.07 | 0.28 | 0.65 | 6.61 | – |
| | $D(x_{\text{MIGA}})$ | 25.91 | 0.86 | 0.15 | 6.74 | 0.83 | 34.84 | 0.94 | 0.14 | 5.05 | 0.75 |

Table 4: Denoising Attack for content replacement and style transfer tasks.

| Network | Image | MAGICBRUSH | | | | | Style Transfer | | | | |
|---------|-------|--------|--------|--------|----------|-----------------|--------|--------|--------|----------|------------------|
| | | PSNR↑ | SSIM↑ | LPIPS↓ | Entropy↓ | CLIP similarity↑ | PSNR↑ | SSIM↑ | LPIPS↓ | Entropy↓ | Gram matrix Loss↓ |
| Xformer | $x$ | ∞ | 1.00 | 0.00 | 7.36 | 0.23 | ∞ | 1.00 | 0.00 | 7.15 | 74.58 |
| | $x_{\text{N}}$ | 34.56 | 0.94 | 0.07 | 7.40 | – | 24.09 | 0.57 | 0.45 | 7.51 | – |
| | $D(x_{\text{N}})$ | 36.60 | 0.97 | 0.05 | 7.36 | 0.23 | 25.56 | 0.79 | 0.34 | 7.03 | 23.75 |
| | $x_{\text{MIGA}}$ | 31.14 | 0.92 | 0.08 | 7.40 | – | 21.76 | 0.52 | 0.45 | 7.52 | – |
| | $D(x_{\text{MIGA}})$ | 26.75 | 0.94 | 0.05 | 7.37 | 0.24 | 21.00 | 0.55 | 0.21 | 7.22 | 9.10 |

## 5.3 ATTACK DENOISING MODELS WITH UNKNOWN TASKS

We evaluate our method's performance on denoising models across tasks with unknown downstream objectives, including text alteration, content replacement, and style transfer, using corresponding reference images. Some results are visualized in Fig. 4. The semantic metrics used are ROUGE-L for text similarity, CLIP for content editing, and Gram matrix loss for style transfer.

**Results on text alteration tasks.** Tab. 3 presents the results for the Tampered-IC13 and Synthetic datasets. We observe that even without knowledge of the downstream task, MIGA can successfully alter the semantic content after denoising while maintaining high image quality. For instance, in the Xformer model on the Tampered-IC13 dataset, the denoised image $D(x_{\text{MIGA}})$ achieves a high PSNR of 25.91 and a low LPIPS of 0.15, while the ROUGE-L score increases to 0.83, indicating significant text alteration.

**Results on content replacement and style transfer.** Tab. 4 shows the results for content replacement using MAGICBRUSH and style transfer tasks. Our method effectively performs semantic modifications in these tasks as well. For example, in the style transfer task, the Gram matrix loss decreases to 9.10 after denoising $D(x_{\text{MIGA}})$, indicating successful style transfer, while maintaining acceptable image quality (Entropy of 7.22).

**Minimal perturbations.** As shown in Tab. 5, the perturbations introduced are minimal, with very low LPIPS (e.g., LPIPS of 0.01 for the Synthetic dataset). This confirms that our method introduces imperceptible changes to the images while disrupting the denoising performance.

Table 5: Perturbation constraint for unknown task.

| Metric | Tampered-IC13 | Synthetic Dataset | MAGICBRUSH | Style Transfer |
|--------|---------------|-------------------|------------|----------------|
| $\text{LPIPS}_{\text{con}}$↓ | 0.02 | 0.01 | 0.01 | 0.04 |

## 5.4 ABLATION STUDY

We perform an ablation study to assess the contributions of different loss components in our method.

**Importance of loss components.**

Tab. 6 summarizes the impact of removing or modifying various loss terms on the Tampered-IC13 dataset using the Xformer model. The key observations are:

*Mutual information loss ($\mathcal{L}_{MI}$):* Removing $\mathcal{L}_{\text{MI}}$ leads to a decrease in the ROUGE-L score from 0.83 to 0.73, indicating less effective semantic alteration. This highlights the importance of $\mathcal{L}_{\text{MI}}$ in influencing semantic alteration during attacking denoising models.

Table 6: Importance of different losses for Tampered-IC13 dataset.

| $L_{\text{con}}$ | | $L_{\text{rec}}$ | | $L_{\text{MI}}$ | Metrics | | | |
|-----|-------------|-----|-------------|-----|-----------------|--------|----------|----------|
| MSE | $L_{\text{perc}}$ | MSE | $L_{\text{perc}}$ | | $\text{LPIPS}_{\text{con}}$↓ | LPIPS↓ | Entropy↓ | ROUGE-L↑ |
| ✓ | ✓ | ✓ | ✓ | ✓ | 0.02 | 0.15 | 6.74 | 0.83 |
| ✓ | ✓ | ✓ | ✓ | ✗ | 0.02 | 0.18 | 6.74 | 0.73 |
| ✓ | ✓ | ✗ | ✗ | ✓ | 0.01 | 0.40 | 6.80 | 0.50 |
| ✗ | ✗ | ✓ | ✓ | ✓ | 0.38 | 0.12 | 6.67 | 0.92 |
| ✓ | ✗ | ✓ | ✗ | ✓ | 0.08 | 0.40 | 6.87 | 0.48 |
| ✗ | ✓ | ✗ | ✓ | ✓ | 0.03 | 0.19 | 6.76 | 0.76 |

***Reconstruction loss ($\mathcal{L}_{rec}$):*** Omitting $\mathcal{L}_{rec}$ increases LPIPS to 0.40, indicating a significant decline in image quality after denoising. This shows that $\mathcal{L}_{rec}$ is crucial for maintaining denoised image clarity.

***Perturbation constraint loss ($\mathcal{L}_{com}$):*** Removing $\mathcal{L}_{com}$ results in the highest ROUGE-L score (0.92) but introduces noticeable perturbations (LPIPS$_{con}$ increases to 0.38), compromising the imperceptibility of the attack.

***Perceptual loss and MSE Loss:*** Using only MSE loss or only perceptual loss adversely affects both image clarity and the effectiveness of semantic alteration. This demonstrates that the combination of loss terms is necessary for optimal performance.

**Effect of task difficulty.** We examine the impact of task difficulty by varying the font size in the Synthetic dataset. As shown in Tab. 7, as the difficulty increases (larger font sizes), there is a slight sacrifice in image clarity (LPIPS increases), but the ROUGE-L score also increases, indicating more effective semantic alteration. Nonetheless, the perturbations remain imperceptible (LPIPS$_{con}$ remains at 0.01).

Table 7: Difficulty evaluation.

| Difficulty | LPIPS$_{con}$ | PSNR↑ | SSIM↑ | LPIPS↓ | Entropy↓ | ROUGE-L↑ |
|---|---|---|---|---|---|---|
| Easy | 0.01 | 34.62 | 0.94 | 0.17 | 5.00 | 0.54 |
| Medium | 0.01 | 34.09 | 0.94 | 0.17 | 5.05 | 0.69 |
| Difficult | 0.01 | 32.55 | 0.93 | 0.19 | 5.21 | 0.73 |

## 6 CONCLUSION

In this work, we propose MIGA, an adversarial attack framework that directly targets image denoising models by disrupting their ability to preserve semantic information. By leveraging mutual information (MI) to quantify semantic consistency, MIGA introduces imperceptible perturbations that selectively alter task-relevant semantics while maintaining visual fidelity. We design three tailored loss functions to optimize this process, enabling attacks in both known and unknown downstream task scenarios. Extensive experiments across multiple denoising models validate MIGA's effectiveness, revealing a critical security risk of denoising models in real-world applications. This work also complements existing adversarial attacks and provides insights into enhancing the robustness of denoising models against semantic manipulation.

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

# Appendix

Our **code** can be found in a separate file under the Supplementary Material, named MIGA-Code.

## A    PROOF OF TASK-RELEVANT MUTUAL INFORMATION REDUCTION

As introduced in Sec. 4.1, our definition of task-relevant mutual information focuses on the semantic features directly relevant to the downstream task, differing from the traditional information-theoretic definition. In this section, we rigorously prove that, when the downstream task model $F$ is known, maximizing the cross-entropy loss $\mathcal{L}_{\text{CE}}(F(D(x_{\text{N}} + \delta)), C)$ is equivalent to minimizing the task-relevant mutual information $I(x; D(x_{\text{N}} + \delta) \mid C)$ between the original image $x$ and the denoised image $D(x_{\text{N}} + \delta)$, conditioned on the ground-truth label $C$.

**Lemma 1.** Increasing the cross-entropy loss $\mathcal{L}_{\text{CE}}(F(D(x_{\text{N}}+\delta)), C)$ reduces the task-relevant mutual information $I(x; D(x_{\text{N}} + \delta) \mid C)$ between the original and denoised images, conditioned on $C$.

*Proof.* We begin by clarifying the notation:

- $x$ denotes the original image.

- $x_{\text{N}}$ represents the noisy version of $x$.

- $D(x_{\text{N}} + \delta)$ is the denoised image after applying a perturbation $\delta$.

- $C$ is the random variable representing the ground-truth class label corresponding to $x$.

- $c$ is a particular class label in the set of all possible classes $\mathcal{C}$.

- $F$ is the downstream classification model, estimating the probability $P_F(c \mid D(x_{\text{N}} + \delta))$ that the denoised image $D(x_{\text{N}} + \delta)$ belongs to class $c$.

Recall the definition of task-relevant mutual information:

$$I(x; D(x_{\text{N}} + \delta) \mid C) = H(C \mid x) - H(C \mid D(x_{\text{N}} + \delta)), \tag{7}$$

where $H(C \mid x)$ and $H(C \mid D(x_{\text{N}} + \delta))$ are the conditional entropies of the label $C$ given the images $x$ and $D(x_{\text{N}} + \delta)$, respectively.

Since $x$ uniquely determines $C$ ( $x$ perfectly predicts $C$), we have $H(C \mid x) = 0$. Thus, the task-relevant mutual information simplifies to:

$$I(x; D(x_{\text{N}} + \delta) \mid C) = -H(C \mid D(x_{\text{N}} + \delta)). \tag{8}$$

Our goal is to minimize $I(x; D(x_{\text{N}} + \delta) \mid C)$, which is equivalent to maximizing $H(C \mid D(x_{\text{N}} + \delta))$.

The conditional entropy $H(C \mid D(x_{\text{N}} + \delta))$ is defined as:

$$H(C \mid D(x_{\text{N}} + \delta)) = -\mathbb{E}_{D(x_{\text{N}}+\delta)} \left[ \sum_{c \in \mathcal{C}} P(c \mid D(x_{\text{N}} + \delta)) \log P(c \mid D(x_{\text{N}} + \delta)) \right], \tag{9}$$

where $P(c \mid D(x_{\text{N}} + \delta))$ is the true probability of class $c$ given the image $D(x_{\text{N}} + \delta)$. The expectation $\mathbb{E}_{D(x_{\text{N}}+\delta)}$ is taken over the distribution of $D(x_{\text{N}} + \delta)$, which represents the probabilistic outputs of the denoising model $D$ given the perturbed input $x_{\text{N}} + \delta$.

In practice, the true conditional probability $P(c \mid D(x_{\text{N}} + \delta))$ is unknown. Instead, we use the high-accuracy and well-calibrated classifier $F$ to estimate it:

$$P(c \mid D(x_{\text{N}} + \delta)) \approx P_F(c \mid D(x_{\text{N}} + \delta)), \tag{10}$$

where $P(c \mid D(x_{\text{N}} + \delta))$ is the probability output by the classifier $F$ for class $c$. The approximation's accuracy depends on the performance of $F$, which we assume is sufficiently high for this analysis.

Substituting this into Equation equation 9, we approximate the conditional entropy:

$$H(C \mid D(x_{\mathrm{N}} + \delta)) \approx -\mathbb{E}_{D(x_{\mathrm{N}}+\delta)}\left[\sum_{c \in \mathcal{C}} P_F(c \mid D(x_{\mathrm{N}} + \delta)) \log P_F(c \mid D(x_{\mathrm{N}} + \delta))\right]. \quad (11)$$

Recall that the cross-entropy loss between the classifier's prediction and the ground-truth label $C$ is:

$$\mathcal{L}_{\mathrm{CE}}(F(D(x_{\mathrm{N}} + \delta)), C) = -\log P_F(C \mid D(x_{\mathrm{N}} + \delta)). \quad (12)$$

Increasing $\mathcal{L}_{\mathrm{CE}}$ is equivalent to reducing $P_F(C \mid D(x_{\mathrm{N}} + \delta))$, i.e., decreasing the classifier's confidence in the correct class $C$. Due to the normalization condition

$$\sum_{c \in \mathcal{C}} P_F(c \mid D(x_{\mathrm{N}} + \delta)) = 1. \quad (13)$$

A decrease in $P_F(C \mid D(x_{\mathrm{N}} + \delta))$ leads to an increase in $P_F(c \mid D(x_{\mathrm{N}} + \delta))$ for $c \neq C$. Assuming no bias or preference for any specific incorrect class, this redistribution results in a more uniform probability distribution $P_F(c \mid D(x_{\mathrm{N}} + \delta))$ closer to the uniform distribution. Consequently, as $P_F(c \mid D(x_{\mathrm{N}} + \delta))$ becomes more uniform, the conditional entropy $H(C \mid D(x_{\mathrm{N}} + \delta))$ increases, consistent with Shannon's entropy theorem Shannon (1948).

Therefore, maximizing $\mathcal{L}_{\mathrm{CE}}$ effectively increases $H(C \mid D(x_{\mathrm{N}} + \delta))$. Substituting back into Equation equation 8, we find that $I(x; D(x_{\mathrm{N}} + \delta) \mid C)$ decreases. Thus, maximizing the cross-entropy loss $\mathcal{L}_{\mathrm{CE}}$ minimizes the task-relevant mutual information $I(x; D(x_{\mathrm{N}} + \delta) \mid C)$. □

## B  TASK-RELEVANT MUTUAL INFORMATION FOR UNKNOWN DOWNSTREAM TASKS

In the main text, we define the task-relevant mutual information $I(x; D(x_{\mathrm{N}} + \delta) \mid C)$ to quantify the shared information between the original image $x$ and the denoised image $D(x_{\mathrm{N}} + \delta)$ that is relevant to a downstream task with ground-truth output $C$.

When the downstream task $F$ is unknown and $C$ is unobservable, directly computing $I(x; D(x_{\mathrm{N}}+\delta) \mid C)$ becomes infeasible. To address this challenge, we propose an approximation strategy that leverages semantically modified images $x_{\mathrm{alt}}$.

Our key assumption is that the semantic alterations introduced in $x_{\mathrm{alt}}$ capture features relevant to potential downstream tasks. By carefully modifying key semantic regions of the original images $x$, we simulate changes that are likely to affect any reasonable downstream task output.

Under this assumption, we approximate the task-relevant mutual information by focusing on the mutual information between $x$ and $D(x_{\mathrm{N}} + \delta)$ with an emphasis on task-relevant features. We achieve this by training a MINE network $T$ Belghazi et al. (2018) using contrastive learning:

- **Positive pairs:** $(x, D(x_{\mathrm{N}}))$, where the denoised image retains the original semantics.
- **Negative pairs:** $(x, x_{\mathrm{alt}})$, where the semantics have been altered.

By maximizing similarity for positive pairs and minimizing it for negative pairs, $T$ learns to focus on task-relevant features. During the attack, we optimize the adversarial perturbation $\delta$ to minimize $T(x, D(x_{\mathrm{N}} + \delta))$, effectively reducing the estimated mutual information related to these features.

This approach allows us to approximate the minimization of $I(x; D(x_{\mathrm{N}} + \delta) \mid C)$ without direct access to $C$. By relying on semantically altered images as proxies for task-relevant changes, we ensure that our MIGA remains effective even when the downstream task is unknown, aligning with our original formulation.

## C  EXPERIMENTAL SETTINGS

### C.1  DATASETS

In this appendix, we provide detailed descriptions of the datasets used in our experiments, focusing on their specific usage and processing steps.

**1.ImageNet-10.** The ImageNet-10 dataset Russakovsky et al. (2015) consists of images from 10 distinct classes. We use this dataset to evaluate classification tasks under noisy conditions. To simulate noise, we add Gaussian noise with a standard deviation $\sigma = 50$ to each image. The images retain their original sizes as provided in the dataset.

**2.Tampered-IC13 dataset.** The Tampered-IC13 dataset Wang et al. (2022) contains images with real-world text alterations in natural scenes, making it suitable for evaluating text tampering detection methods. We apply OCR to detect and recognize text regions within each image. The original images vary in size. For each detected text region, we extract a crop of size $256 \times 256$ pixels centered on the text location to standardize the input size. We then add Gaussian noise with $\sigma = 25$ to each cropped image, resulting in a total of 404 noisy images.

**3.MAGICBRUSH dataset.** The MAGICBRUSH dataset Zhang et al. (2024) is a large-scale, manually annotated, instruction-guided image editing dataset. It covers diverse scenarios, including single-turn and multi-turn edits, as well as mask-provided and mask-free editing tasks. The dataset contains 10,000 (source image, instruction, target image) triples, making it ideal for training and evaluating image editing models. We resize all original images, which vary in size, to $256 \times 256$ pixels to ensure consistency. Gaussian noise with $\sigma = 25$ is added to the source images.

**4.Neural style Transfer dataset.** This synthetic dataset is designed to evaluate models on stylized images. We select images from the DFWB datasets (DIV2K Agustsson & Timofte (2017), Flickr2K, WED Ma et al. (2016), BSD Martin et al. (2001)) and resize them to $256 \times 256$ pixels. Gaussian noise with $\sigma = 25$ is added to these images to serve as noisy inputs. We use the neural style transfer method from Gatys et al. Gatys (2015), training the style transfer model for 500 epochs to generate stylized versions of the images. These stylized images serve as the target images, resulting in a total of 2,921 image pairs.

**5.Synthetic text Alteration dataset.** We create this dataset to evaluate models on text alteration tasks involving small but semantically significant changes. We select 3,000 patches from the SIDD dataset Abdelhamed et al. (2018), each resized to $256 \times 256$ pixels. Two sets of text overlays, containing five different digits and letters, are placed at the same location within each image patch. One set is corrupted with Gaussian noise ($\sigma = 25$) to serve as the noisy original images, while the other set remains clean to serve as the target images.

In our experiments, we add Gaussian noise to images across different datasets to create noisy versions. Specifically, Gaussian noise with $\sigma = 50$ is applied to the ImageNet-10 dataset, while Gaussian noise with $\sigma = 25$ is used for all other datasets. When computing the reconstruction loss $\mathcal{L}_{\text{rec}}$, we use the clean image $x_{\text{ref}}$ as the reference without any added noise. Some visualization results of the clean aforementioned dataset are shown in Fig. 5.

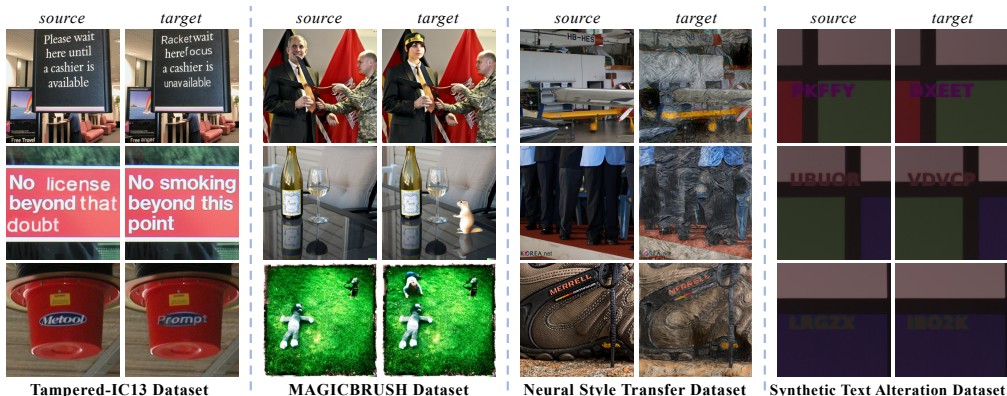

Figure 5: Example images from our datasets, including a pair of related images with differing semantics.

## C.2 Evaluation metrics

To thoroughly evaluate the performance of our algorithm, we employ a comprehensive set of metrics focusing on three key aspects: imperceptibility of the perturbations, clarity of the denoised images, and semantic fidelity with respect to downstream tasks.

### C.2.1 Imperceptibility measures

To assess the imperceptibility of the adversarial perturbations, we compute the following metrics between the noisy image $x_N$ and the adversarial image $x_N + \delta$:

- **PSNR$_{con}$:** It measures the pixel-wise similarity, with higher values indicating smaller differences and thus less perceptible perturbations Wang & Bovik (2002).
- **SSIM$_{con}$:** It evaluates structural similarity, with values closer to 1 indicating that the perturbations are structurally imperceptible Wang & Bovik (2002).
- **LPIPS$_{con}$:** It quantifies perceptual similarity, with lower scores indicating higher perceptual similarity and thus more imperceptible perturbations Zhang et al. (2018).
- **Entropy$_{con}$:** It measures the difference in image entropy between $x_N$ and $x_N + \delta$. Minimal changes in entropy indicate that the perturbations do not introduce noticeable randomness or complexity, helping to keep the perturbations imperceptible Shannon (1948).

These imperceptibility metrics, denoted with the subscript $_{con}$, ensure that the adversarial perturbations $\delta$ added to the noisy images are imperceptible to human observers, thus maintaining the stealthiness of the attack.

### C.2.2 Distortion and clarity measures

To evaluate the clarity and quality of the denoised images, we compute the following metrics between the denoised images and the original clean images:

- **PSNR:** It measures the pixel-wise similarity between the original image $x$ and the denoised images. Higher PSNR values indicate better denoising quality Wang & Bovik (2002).
- **SSIM:** It assesses the structural similarity between the original and denoised images, with values closer to 1 indicating higher structural fidelity Wang & Bovik (2002).
- **LPIPS:** It computes perceptual similarity between the original image $x$ and the denoised images, with lower scores indicating better perceptual quality Zhang et al. (2018).
- **Entropy:** It evaluates the amount of information or randomness in the denoised images. Appropriate levels of entropy indicate the preservation of image details without introducing artifacts. An excessively high entropy may indicate residual noise or artifacts, while an entropy value that is too low may suggest over-smoothing and loss of important details. Shannon (1948).

These metrics are calculated using clean reference images, $x_{ref}$, as the benchmark. Our objective is to ensure that the denoised outputs, $D(x_N)$ and $D(x_N + \delta)$, exhibit high quality and maintain a close resemblance to the corresponding clean images. This demonstrates the efficacy of the denoising process, even when faced with adversarial perturbations.

### C.2.3 Semantic fidelity measures

To assess the impact of adversarial perturbations on semantic alterations, we employ task-specific performance metrics of downstream tasks, providing detailed evaluations for each scenario.

- **Classification tasks:** For evaluating classification tasks, we use a pretrained downstream classifier $F$ to classify the denoised adversarial images $D(x_N + \delta)$. We measure the classification accuracy and compare it with that of the clean images $x$ and the denoised images $D(x_N)$. A significant decrease in accuracy on $D(x_N + \delta)$ indicates that the adversarial perturbations have successfully altered the semantic content of the denoised images, causing

the classifier to misclassify them. This demonstrates the effectiveness of the attack in compromising the semantic fidelity of denoising models.

- **Text alteration:** In tasks involving text recognition, such as Optical Character Recognition (OCR) Singh et al. (2012), we employ the ROUGE-L metric Lin (2004) to evaluate the semantic changes introduced by adversarial perturbations. This metric measures the Longest Common Subsequence (LCS) between the original text and the OCR results obtained from the reference image (which has been semantically modified), capturing differences in lexical choices and sentence structure. A higher ROUGE-L score indicates that the adversarial perturbations have successfully caused significant semantic deviations, demonstrating their effectiveness in altering the textual content during denoising.

- **Image modification:** For tasks where images are supposed to be modified according to textual instructions (e.g., adding an object, changing colors), we evaluate how well the denoised adversarial image $D(x_{\mathrm{N}} + \delta)$ aligns with the intended modification instructions. We employ Contrastive Language–Image Pre-training (CLIP) similarity Radford et al. (2021), which computes the cosine similarity between the image and text embeddings in a shared multimodal space. CLIP has been shown to effectively capture semantic relationships between images and text. A higher CLIP similarity score between $D(x_{\mathrm{N}} + \delta)$ and the modification instructions indicates that the adversarial perturbations have successfully enabled the denoised image to adhere to the intended semantic modifications, thereby demonstrating the effectiveness of the attack.

- **Style transfer:** In style transfer tasks, we assess the impact of adversarial perturbations on the stylistic alignment between the denoised adversarial image $D(x_{\mathrm{N}} + \delta)$ and the target style image. We compute the Gram matrix loss Gatys (2015), which measures the difference in style by comparing the correlations between feature maps (Gram matrices) of the two images extracted from a convolutional neural network's layers. The Gram matrix captures the texture and style information of an image. A lower Gram matrix loss indicates better alignment with the target style, suggesting that the adversarial perturbations have successfully enabled the denoised image to achieve the intended style transfer, thereby demonstrating their effectiveness.

### C.2.4 OVERALL EVALUATION

Our comprehensive evaluation balances imperceptibility, clarity, and semantic impact. By ensuring that the adversarial perturbations are imperceptible ($\text{PSNR}_{\text{con}}$, $\text{SSIM}_{\text{con}}$, $\text{LPIPS}_{\text{con}}$, and $\text{Entropy}_{\text{con}}$), we maintain the stealthiness of the attack. Simultaneously, high clarity metrics (PSNR, SSIM, LPIPS, and Entropy) between the denoised images and the original image confirm the effectiveness of the denoising process. Finally, semantic fidelity quantifies semantic modifications during the denoising process through downstream task evaluations, demonstrating the practical implications of our adversarial perturbations.

### C.3 IMPLEMENTATION DETAILS

### C.3.1 HYPERPARAMETER

All experiments were conducted using NVIDIA V100 GPUs (32GB). Unlike traditional $\ell_p$-bounded attacks, MIGDA's loss-driven approach does not require $\epsilon$ or $\alpha$. It typically converges within 30 iterations, taking approximately $0.8$ seconds on a $256 \times 256$ image. The Adam optimizer Diederik (2014) with an initial learning rate of $10^{-4}$ was employed during the attack phase. We employed a learning rate scheduler with a step size of 30 and a decay factor ($\gamma = 0.5$) to dynamically adjust the learning rate during training. Through an extensive grid search process, the optimal set of hyperparameters was determined to be $\alpha = 1$, $\beta = 0.1$, $\lambda_{\text{con}} = 3$, $\lambda_{\text{rec}} = 1$, and $\lambda_{\text{MI}} = 0.01$. Detailed experimental results and further discussion of these choices are presented in Appendix D.3.

### C.3.2 TRAINING OF MINE NETWORK

For unknown downstream tasks, we used the MINE network to estimate the task-related mutual information. Specifically, we trained a neural network estimator $T$ based on the MINE framework Belghazi et al. (2018), where the positive sample pair consists of the original image $x$ and the denoised

Table 8: Denoising attack for text alteration tasks.

| Network | Image | Tampered-IC13 | | | | | Synthetic Dataset | | | | |
|---|---|---|---|---|---|---|---|---|---|---|---|
| | | PSNR↑ | SSIM↑ | LPIPS↓ | Entropy↓ | ROUGE-L↑ | PSNR↑ | SSIM↑ | LPIPS↓ | Entropy↓ | ROUGE-L↑ |
| | $x$ | ∞ | 1.00 | 0.00 | 6.60 | 0.57 | ∞ | 1.00 | 0.00 | 5.05 | 0.15 |
| | $x_{\mathrm{N}}$ | 23.98 | 0.47 | 0.48 | 7.32 | – | 24.15 | 0.28 | 0.65 | 6.61 | – |
| Restormer | $D(x_{\mathrm{N}})$ | 27.89 | 0.87 | 0.29 | 6.65 | 0.57 | 28.22 | 0.89 | 0.42 | 5.28 | 0.15 |
| | $x_{\mathrm{adv}}$ | 25.58 | 0.56 | 0.46 | 7.05 | – | 26.12 | 0.42 | 0.63 | 6.23 | – |
| | $D(x_{\mathrm{adv}})$ | 13.92 | 0.38 | 0.58 | 6.52 | 0.32 | 17.65 | 0.62 | 0.63 | 5.44 | 0.01 |
| | $x_{\mathrm{MIGA}}$ | 23.06 | 0.45 | 0.48 | 7.32 | – | 24.32 | 0.29 | 0.62 | 6.58 | – |
| | $D(x_{\mathrm{MIGA}})$ | 24.95 | 0.85 | 0.17 | 6.75 | 0.77 | 34.09 | 0.94 | 0.17 | 5.05 | 0.69 |
| Xformer | $D(x_{\mathrm{N}})$ | 27.57 | 0.85 | 0.30 | 6.61 | 0.57 | 31.70 | 0.90 | 0.39 | 5.10 | 0.15 |
| | $x_{\mathrm{adv}}$ | 26.13 | 0.56 | 0.46 | 7.05 | – | 26.54 | 0.42 | 0.62 | 6.23 | – |
| | $D(x_{\mathrm{adv}})$ | 13.79 | 0.37 | 0.68 | 6.47 | 0.28 | 16.98 | 0.61 | 0.68 | 5.29 | 0.00 |
| | $x_{\mathrm{MIGA}}$ | 22.92 | 0.45 | 0.48 | 7.34 | – | 24.07 | 0.28 | 0.65 | 6.61 | – |
| | $D(x_{\mathrm{MIGA}})$ | 25.91 | 0.86 | 0.15 | 6.74 | 0.83 | 34.84 | 0.94 | 0.14 | 5.05 | 0.75 |
| PromptIR | $D(x_{\mathrm{N}})$ | 27.89 | 0.87 | 0.23 | 6.65 | 0.57 | 38.42 | 0.95 | 0.27 | 5.06 | 0.15 |
| | $x_{\mathrm{adv}}$ | 23.88 | 0.50 | 0.49 | 7.21 | – | 24.11 | 0.35 | 0.66 | 6.49 | – |
| | $D(x_{\mathrm{adv}})$ | 14.30 | 0.26 | 0.63 | 7.07 | 0.35 | 20.17 | 0.49 | 0.65 | 6.25 | 0.01 |
| | $x_{\mathrm{MIGA}}$ | 22.33 | 0.43 | 0.49 | 7.34 | – | 23.56 | 0.27 | 0.65 | 6.63 | – |
| | $D(x_{\mathrm{MIGA}})$ | 23.65 | 0.83 | 0.25 | 6.81 | 0.77 | 29.74 | 0.86 | 0.28 | 5.05 | 0.54 |
| AFM | $D(x_{\mathrm{N}})$ | 26.83 | 0.80 | 0.38 | 6.72 | 0.57 | 32.53 | 0.87 | 0.21 | 5.41 | 0.15 |
| | $x_{\mathrm{adv}}$ | 24.48 | 0.53 | 0.48 | 7.32 | – | 26.22 | 0.43 | 0.64 | 6.25 | – |
| | $D(x_{\mathrm{adv}})$ | 13.98 | 0.31 | 0.65 | 7.12 | 0.31 | 17.32 | 0.64 | 0.69 | 5.52 | 0.01 |
| | $x_{\mathrm{MIGA}}$ | 22.76 | 0.44 | 0.48 | 7.35 | – | 25.47 | 0.32 | 0.67 | 6.62 | – |
| | $D(x_{\mathrm{MIGA}})$ | 25.48 | 0.84 | 0.18 | 6.78 | 0.83 | 35.72 | 0.93 | 0.15 | 5.27 | 0.72 |

Table 9: Denoising attack for content replacement and style transfer tasks.

| Network | Image | MAGICBRUSH | | | | | Style Transfer | | | | |
|---|---|---|---|---|---|---|---|---|---|---|---|
| | | PSNR↑ | SSIM↑ | LPIPS↓ | Entropy↓ | CLIP similarity↑ | PSNR↑ | SSIM↑ | LPIPS↓ | Entropy↓ | Gram matrix Loss↓ |
| | $x$ | ∞ | 1.00 | 0.00 | 7.36 | 0.23 | ∞ | 1.00 | 0.00 | 7.15 | 74.58 |
| | $x_{\mathrm{N}}$ | 34.56 | 0.94 | 0.07 | 7.40 | – | 24.09 | 0.57 | 0.45 | 7.51 | – |
| Restormer | $D(x_{\mathrm{N}})$ | 37.43 | 0.97 | 0.05 | 7.37 | 0.23 | 26.57 | 0.83 | 0.32 | 7.09 | 21.98 |
| | $x_{\mathrm{adv}}$ | 25.26 | 0.62 | 0.45 | 7.46 | – | 25.24 | 0.62 | 0.46 | 7.27 | – |
| | $D(x_{\mathrm{adv}})$ | 13.22 | 0.28 | 0.63 | 7.16 | 0.22 | 12.57 | 0.16 | 0.68 | 6.89 | 327.53 |
| | $x_{\mathrm{MIGA}}$ | 31.64 | 0.93 | 0.08 | 7.40 | – | 22.41 | 0.52 | 0.45 | 7.50 | – |
| | $D(x_{\mathrm{MIGA}})$ | 26.07 | 0.93 | 0.05 | 7.38 | 0.24 | 21.45 | 0.56 | 0.25 | 7.31 | 13.95 |
| Xformer | $D(x_{\mathrm{N}})$ | 36.60 | 0.97 | 0.05 | 7.36 | 0.23 | 25.56 | 0.79 | 0.34 | 7.03 | 23.75 |
| | $x_{\mathrm{adv}}$ | 25.91 | 0.63 | 0.44 | 7.47 | – | 25.91 | 0.63 | 0.45 | 7.28 | – |
| | $D(x_{\mathrm{adv}})$ | 13.31 | 0.28 | 0.72 | 6.97 | 0.22 | 13.16 | 0.16 | 0.73 | 6.74 | 418.85 |
| | $x_{\mathrm{MIGA}}$ | 31.14 | 0.92 | 0.08 | 7.40 | – | 21.76 | 0.52 | 0.45 | 7.52 | – |
| | $D(x_{\mathrm{MIGA}})$ | 26.75 | 0.94 | 0.05 | 7.37 | 0.24 | 21.00 | 0.55 | 0.21 | 7.22 | 9.10 |
| PromptIR | $D(x_{\mathrm{N}})$ | 42.27 | 0.99 | 0.03 | 7.36 | 0.23 | 30.13 | 0.91 | 0.21 | 7.13 | 110.77 |
| | $x_{\mathrm{adv}}$ | 23.48 | 0.55 | 0.48 | 7.57 | – | 23.56 | 0.55 | 0.48 | 7.40 | – |
| | $D(x_{\mathrm{adv}})$ | 12.98 | 0.23 | 0.62 | 7.37 | 0.22 | 12.85 | 0.12 | 0.66 | 7.33 | 1467.96 |
| | $x_{\mathrm{MIGA}}$ | 30.00 | 0.91 | 0.09 | 7.40 | – | 21.40 | 0.49 | 0.47 | 7.48 | – |
| | $D(x_{\mathrm{MIGA}})$ | 25.65 | 0.93 | 0.05 | 7.38 | 0.24 | 20.78 | 0.59 | 0.28 | 7.42 | 52.88 |
| AFM | $D(x_{\mathrm{N}})$ | 36.21 | 0.95 | 0.06 | 7.35 | 0.23 | 25.53 | 0.71 | 0.33 | 7.09 | 23.32 |
| | $x_{\mathrm{adv}}$ | 25.87 | 0.62 | 0.45 | 7.47 | – | 24.33 | 0.63 | 0.47 | 7.28 | – |
| | $D(x_{\mathrm{adv}})$ | 13.33 | 0.29 | 0.70 | 7.08 | 0.22 | 13.22 | 0.17 | 0.69 | 6.73 | 455.92 |
| | $x_{\mathrm{MIGA}}$ | 30.69 | 0.92 | 0.07 | 7.40 | – | 21.29 | 0.49 | 0.45 | 7.54 | – |
| | $D(x_{\mathrm{MIGA}})$ | 27.10 | 0.93 | 0.08 | 7.37 | 0.24 | 20.15 | 0.51 | 0.25 | 7.26 | 15.34 |

image $D(x_{\mathrm{N}})$, and the negative sample pair consists of the original image $x$ and a semantically altered image $x_{\mathrm{alt}}$. The goal is to let the network $T$ distinguish between positive and negative sample pairs to estimate the mutual information value for a given image pair.

The training process uses a contrastive loss function:

$$L_{\mathrm{contrastive}} = -\log\left(\frac{\exp(T(x, D(x_{\mathrm{N}})))}{\exp(T(x, D(x_{\mathrm{N}}))) + \exp(T(x, x_{\mathrm{alt}}))}\right). \quad (14)$$

This loss function encourages the network to assign higher similarity scores to positive sample pairs, while giving lower scores to negative sample pairs. After training, $T(x, D(x_{\mathrm{N}}))$ represents the similarity between the original image $x$ and the denoised image, reflecting the semantic relevance of the image.

# D EXPERIMENTAL RESULTS

## D.1 DENOISING ATTACK FOR UNKNOWN DOWNSTREAM TASKS

In this section, we further evaluate the performance of MIGA in scenarios where the downstream tasks are unknown to the attacker. Tab. 8 and Tab.9 present a comprehensive comparison between our

Table 10: Perturbation constraint results. The values before the slash represent the constraint magnitude of traditional I-FGSM attack methods compared to the original images, while the values after the slash represent the constraint magnitude of MIGA.

| Model | Metric | ImageNet-10 | Tampered-IC13 | Synthetic Dataset | MAGICBRUSH | Style Transfer |
|-------|--------|-------------|---------------|-------------------|------------|----------------|
| Restormer | $PSNR_{con}\uparrow$ | 30.38/38.53 | 31.26/28.11 | 31.75/35.66 | 30.96/33.82 | 30.80/25.87 |
| | $SSIM_{con}\uparrow$ | 0.98/0.99 | 0.95/0.92 | 0.93/0.94 | 0.94/0.97 | 0.94/0.87 |
| | $LPIPS_{con}\downarrow$ | 0.04/0.01 | 0.07/0.02 | 0.11/0.01 | 0.08/0.01 | 0.08/0.04 |
| Xformer | $PSNR_{con}\uparrow$ | 30.51/41.56 | 32.50/29.00 | 33.21/35.66 | 32.31/33.34 | 32.16/25.10 |
| | $SSIM_{con}\uparrow$ | 0.98/0.99 | 0.94/0.94 | 0.93/0.97 | 0.88/0.97 | 0.94/0.88 |
| | $LPIPS_{con}\downarrow$ | 0.04/0.01 | 0.08/0.02 | 0.12/0.01 | 0.08/0.01 | 0.08/0.04 |
| PromptIR | $PSNR_{con}\uparrow$ | 28.97/34.39 | 28.46/26.65 | 28.01/30.30 | 27.58/31.40 | 27.74/23.51 |
| | $SSIM_{con}\uparrow$ | 0.97/0.99 | 0.92/0.90 | 0.89/0.91 | 0.89/0.96 | 0.89/0.82 |
| | $LPIPS_{con}\downarrow$ | 0.07/0.01 | 0.15/0.04 | 0.27/0.03 | 0.18/0.02 | 0.18/0.08 |
| AFM | $PSNR_{con}\uparrow$ | 31.48/39.82 | 30.02/28.16 | 32.24/34.47 | 27.58/28.26 | 28.64/24.49 |
| | $SSIM_{con}\uparrow$ | 0.98/0.99 | 0.93/0.92 | 0.92/0.96 | 0.87/0.94 | 0.91/0.85 |
| | $LPIPS_{con}\downarrow$ | 0.02/0.01 | 0.09/0.02 | 0.11/0.01 | 0.19/0.02 | 0.12/0.03 |

Figure 6: Attack results on unknown tasks. Traditional attacks like I-FGSM degrade image quality, while MIGA modifies key semantic information in the denoised images generated after the attack.

proposed method MIGA and the traditional adversarial attack method I-FGSM across multiple state-of-the-art denoising models, including Restormer, Xformer, PromptIR, and AFM. The evaluation is conducted on several datasets, namely Tampered-IC13, Synthetic Dataset, MAGICBRUSH, and Style Transfer tasks, to ensure the generality of our findings. Tab. 10 quantifies the degree of perturbations introduced to the original images by both methods. Specifically, it reports the perceptual similarity metrics such as $PSNR_{con}$, $SSIM_{con}$, and $LPIPS_{con}$ between the adversarial examples and the original noisy images. Additionally, Fig. 6 provides visual illustrations of the denoised images after the attacks, highlighting the perceptual quality and the effectiveness of the perturbations in altering the semantics of the denoised image.

Consistent with the conclusions drawn from Sec. 5.3, our results demonstrate that MIGA effectively introduces imperceptible perturbations that remain undetectable to the human eye while significantly impacting the performance of downstream tasks after denoising. For instance, when using the Restormer network on the Tampered-IC13 dataset, the perceptual metric $LPIPS_{con}$ increases only marginally by 0.02, indicating minimal perceptual difference from the original image. However, the ROUGE-L score, which measures the quality of text recognition or generation tasks, significantly improves to 0.77, reflecting the success of the attack in altering the downstream task outcome. In contrast, images attacked by I-FGSM show a substantial decrease in PSNR and SSIM values, leading to noticeable degradation in image quality. This degradation not only makes the perturbations perceptible but also could raise suspicion in real-world applications where image integrity is crucial. Our method, therefore, presents a more stealthy and effective approach for attacking denoising models in scenarios where the attacker lacks knowledge of the specific downstream tasks.

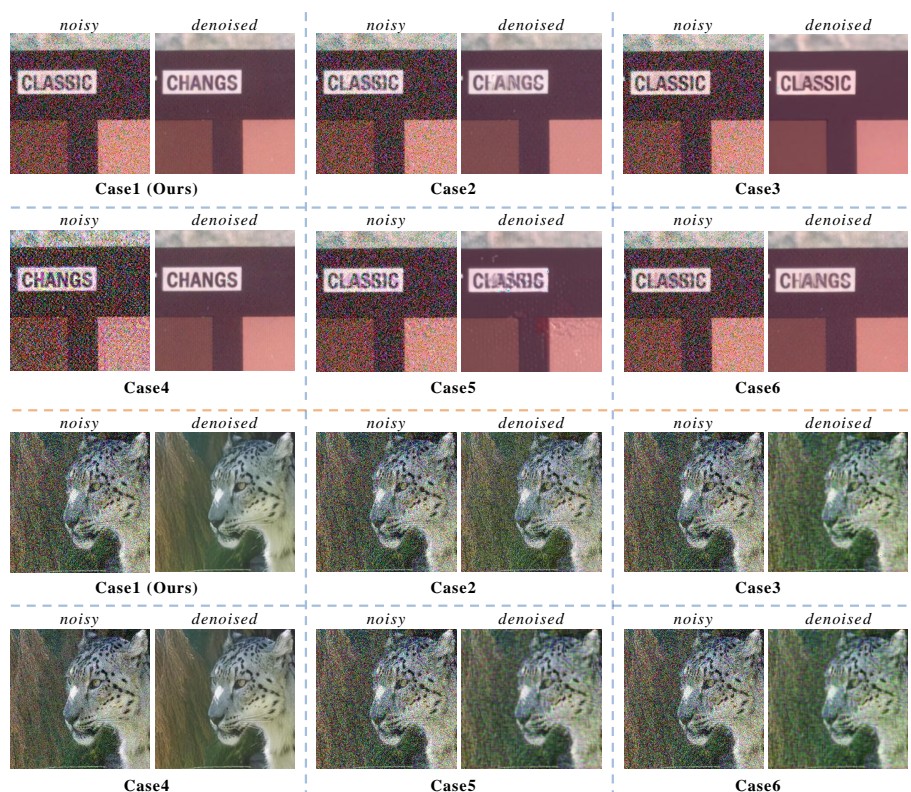

Figure 7: Attack results of MIGA on the Tampered-IC13 (top) and ImageNet-10 (bottom) datasets under different loss combinations.Specifically, *case1* through *case6* sequentially correspond to the experimental settings listed from top to bottom in Tab. 11.

Table 11: Importance of different losses for known task.

| $L$com | | $L$rec | | $L_{\mathrm{MI}}$ | Metrics | | | |
|---|---|---|---|---|---|---|---|---|
| MSE | $L$perc | MSE | $L$perc | | LPIPS(con) | LPIPS | Entropy | Accuracy |
| ✓ | ✓ | ✓ | ✓ | ✓ | 0.01 | 0.26 | 7.06 | 51.54 |
| ✓ | ✓ | ✓ | ✓ | × | 0.01 | 0.26 | 7.36 | 86.04 |
| ✓ | ✓ | × | × | ✓ | 0.01 | 0.37 | 7.41 | 52.08 |
| × | × | ✓ | ✓ | ✓ | 0.01 | 0.36 | 7.41 | 51.46 |
| ✓ | × | ✓ | × | ✓ | 0.01 | 0.36 | 7.40 | 52.19 |
| × | ✓ | × | ✓ | ✓ | 0.01 | 0.36 | 7.41 | 51.58 |

## D.2 IMPORTANCE OF LOSS COMPONENTS

To further understand the contribution of each component in our loss function design, we add ablation studies under the scenario where the downstream tasks are known. The experimental results are detailed in Tab. 11, which lists various combinations of the loss components and their corresponding performance metrics. Fig. 7 visually compares the outcomes under different ablation settings, providing qualitative insights into how each loss term affects the final results. Specifically, *case1* through *case6* sequentially correspond to the experimental settings listed from top to bottom in Tab. 11. Our comprehensive analysis reveals that the inclusion of all loss components in our method yields the best performance. Specifically, incorporating both the content loss ($L_{\mathrm{con}}$) and the reconstruction loss ($L_{\mathrm{rec}}$) alongside the mutual information loss ($L_{\mathrm{MI}}$) ensures that the adversarial perturbations are minimal, the denoised images are clear, and the semantic content is effectively altered.

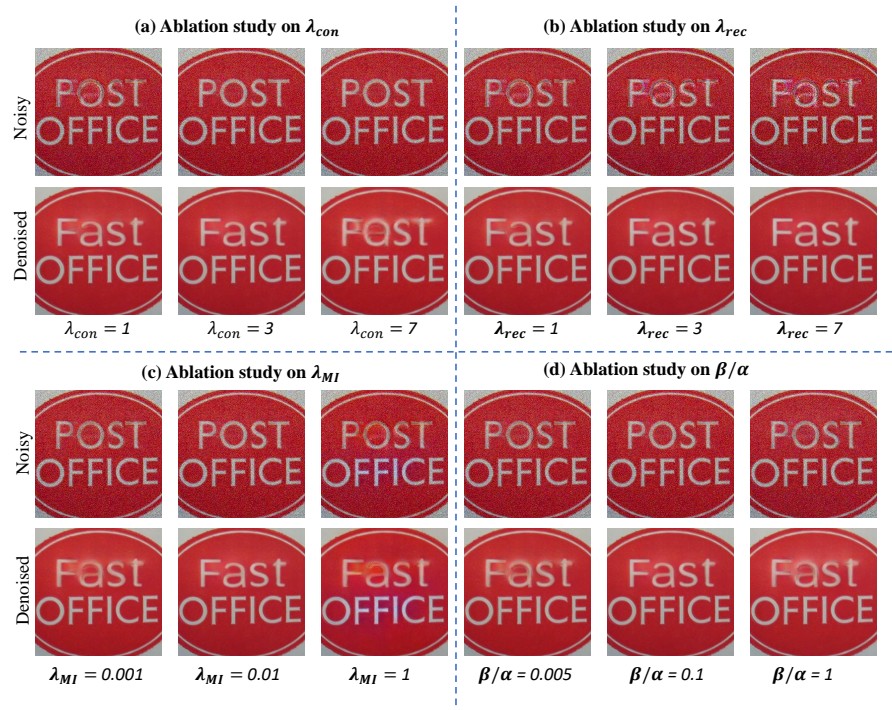

Figure 8: Hyperparameter selection results.

Notably, when certain loss components are omitted, the performance degrades. For example, *case4*, which excludes the content loss, results in clear denoised images but lacks constraints on the perturbations, leading to significant alterations in the original images. This is evident in the Tampered-IC13 dataset results, where semantic content is visibly changed even before denoising. Such changes could be easily detected, compromising the stealthiness of the attack. Therefore, our loss function design is crucial for balancing the imperceptibility of the perturbations with the effectiveness of the attack.

## D.3 HYPERPARAMETER SELECTION

The selection of hyperparameters in our algorithm plays a pivotal role in balancing the trade-offs between perturbation imperceptibility, denoised image clarity, and semantic alteration of the downstream tasks. The hyperparameters $\alpha$, $\beta$, $\lambda_{\text{con}}$, $\lambda_{\text{rec}}$, and $\lambda_{\text{MI}}$ correspond to the weights of different loss components and constraints in our optimization objective. As discussed in Appendix D.2, $\lambda_{\text{con}}$ regulates the magnitude of the perturbations added to the original images, ensuring that they remain imperceptible. $\lambda_{\text{rec}}$ influences the reconstruction fidelity of the denoised images, affecting their visual clarity. $\lambda_{\text{MI}}$ controls the degree to which the mutual information between the denoised images and the original images is minimized, thereby altering the semantic content relevant to the downstream tasks. The ratio of $\alpha$ to $\beta$ determines the balance between pixel-level fidelity and perceptual similarity in the constraint loss.

To empirically determine the optimal hyperparameters, we performed a grid search over a range of values, systematically evaluating the impact of each parameter on the attack's effectiveness and imperceptibility. Fig. 8 presents a subset of the results from our hyperparameter tuning experiments. Our analysis indicates that the optimal settings are $\alpha = 1$, $\beta = 0.1$, $\lambda_{\text{con}} = 3$, $\lambda_{\text{rec}} = 1$, and $\lambda_{\text{MI}} = 0.01$. These values achieve a harmonious balance, ensuring that the perturbations are imperceptible, the denoised images are visually clear, and the downstream tasks are effectively disrupted. These settings were consistently effective across different datasets and denoising models, demonstrating the robustness of our approach.

Table 12: Comparison of different adversarial attacks on the Tampered-IC13 dataset.

| Network | Image | PSNR↑ | SSIM↑ | LPIPS↓ | Entropy↓ | ROUGE-L↑ |
|---|---|---|---|---|---|---|
| | $x$ | $\infty$ | 1.00 | 0.00 | 6.60 | 0.57 |
| | $x_{\mathrm{N}}$ | 23.98 | 0.47 | 0.48 | 7.32 | – |
| | $D(x_{\mathrm{N}})$ | 27.57 | 0.85 | 0.30 | 6.61 | 0.57 |
| | $x_{\mathrm{adv}}$ | 26.13 | 0.56 | 0.46 | 7.05 | – |
| | $D(x_{\mathrm{adv}})$ | 13.79 | 0.37 | 0.68 | 6.47 | 0.28 |
| Xformer | $x_{\mathrm{PGD}}$ | 22.32 | 0.42 | 0.48 | 7.09 | – |
| | $D(x_{\mathrm{PGD}})$ | 17.09 | 0.40 | 0.53 | 6.61 | 0.02 |
| | $x_{\mathrm{AutoAttack})}$ | 22.31 | 0.42 | 0.45 | 7.21 | – |
| | $D(x_{\mathrm{AutoAttack}})$ | 17.68 | 0.44 | 0.50 | 6.61 | 0.02 |
| | $x_{\mathrm{CosPGD(original)}}$ | 27.22 | 0.60 | 0.42 | 7.03 | – |
| | $D(x_{\mathrm{CosPGD(original)}})$ | 14.94 | 0.41 | 0.52 | 6.48 | 0.01 |
| | $x_{\mathrm{CosPGD(adapted)}}$ | 24.06 | 0.48 | 0.46 | 7.29 | – |
| | $D(x_{\mathrm{CosPGD(adapted)}})$ | 22.80 | 0.78 | 0.28 | 6.52 | 0.12 |
| | $x_{\mathrm{MIGA}}$ | 22.92 | 0.45 | 0.48 | 7.34 | – |
| | $D(x_{\mathrm{MIGA}})$ | 25.91 | 0.86 | 0.15 | 6.74 | 0.83 |

Table 13: Comparison of perturbation constraints on the Tampered-IC13 task.

| Metric | I-FGSM | PGD | AutoAttack | CosPGD(original) | CosPGD(adapted) | MIGA |
|---|---|---|---|---|---|---|
| $\mathrm{LPIPS_{con}}$↓ | 0.04 | 0.01 | 0.01 | 0.01 | 0.02 | 0.01 |

## D.4 RESULTS OF OTHER ADVANCED ADVERSARIAL ATTACK METHODS

We extend our evaluation by comparing MIGA with several state-of-the-art adversarial attacks, including CosPGD Agnihotri et al. (2023), PGD Madry et al. (2017), and AutoAttack Croce & Hein (2020). Among them, CosPGD is a pixel-level attack designed primarily for degrading image quality. While effective at reducing image clarity, it fails to manipulate high-level semantics when applied to denoising models. To further explore its potential for semantic attacks, we adapt CosPGD by introducing semantically modified target images. As shown in Tab. 12, the original CosPGD substantially lowers PSNR without altering semantics, whereas the adapted version maintains better image clarity but still struggles to achieve meaningful semantic shifts.

We additionally compare PGD and AutoAttack, both of which are widely used gradient-based adversarial methods. Similar to CosPGD, these attacks degrade image quality (e.g., PSNR and SSIM) and slightly affect downstream outputs. However, they are largely ineffective in inducing targeted semantic manipulations, achieving only ROUGE-L scores of 0.02 after denoising, which is comparable to the failure rate of CosPGD. In contrast, MIGA explicitly targets semantic consistency while preserving visual fidelity, achieving a more deceptive and effective attack. For instance, MIGA achieves a ROUGE-L score of 0.83, significantly outperforming all other methods in semantic manipulation. Finally, Tab. 13 confirms that all attack methods—including PGD, AutoAttack, and the adapted CosPGD—introduce imperceptible perturbations as measured by LPIPS, ensuring a fair comparison under similar perceptual constraints.

Table 14: Transfer evaluation: Xformer-Restormer represents adversarial samples generated against Xformer and tested on Restormer.

| Transfer | PSNR↑ | SSIM↑ | LPIPS↓ | Entropy↓ | Accuracy↓ |
|---|---|---|---|---|---|
| Origin | 17.65 | 0.30 | 0.62 | 7.71 | 83.00 |
| No-Attack | 22.36 | 0.70 | 0.38 | 6.81 | 89.85 |
| Xformer-Xformer | 25.64 | 0.67 | 0.22 | 7.13 | 35.23 |
| Xformer-Restormer | 24.95 | 0.77 | 0.32 | 7.36 | 35.92 |
| PromptIR-PromptIR | 24.20 | 0.59 | 0.37 | 7.60 | 21.31 |
| PromptIR-Restormer | 24.94 | 0.66 | 0.38 | 7.36 | 31.04 |

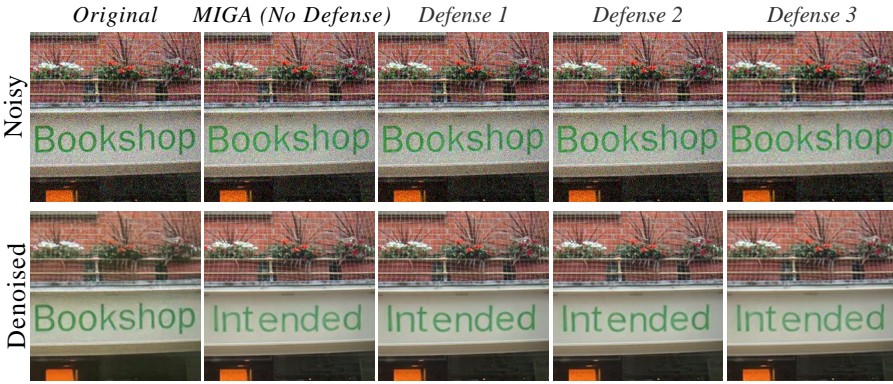

Figure 9: Impact of different defense strategies on MIGA. A comparison of feature squeezing (Defense 1), non-local means denoising (Defense 2), and DiffPure (Defense 3) on image denoising.

Table 15: Robustness evaluation

| Defense | PSNR↑ | SSIM↑ | LPIPS↓ | Entropy↓ | ROUGE-L↑ |
|---|---|---|---|---|---|
| Origin | 23.98 | 0.47 | 0.48 | 7.32 | – |
| No-Defense | 25.91 | 0.86 | 0.15 | 6.74 | 0.83 |
| Feature Squeezing | 26.04 | 0.87 | 0.13 | 6.73 | 0.80 |
| Non Local Means Denoising | 25.53 | 0.87 | 0.13 | 6.74 | 0.80 |
| DiffPure | 25.92 | 0.85 | 0.16 | 6.73 | 0.77 |

## D.5    TRANSFERABILITY ON OTHER DENOISING MODELS

An important aspect of adversarial attacks is their transferability—the ability of adversarial examples generated for one model to be effective against other models. To evaluate this, we tested the adversarial examples generated using one denoising model on other models, examining both the degradation in downstream task performance and the preservation of image quality. As summarized in Tab. 14, our adversarial examples exhibit significant transferability across different denoising architectures. For instance, adversarial examples crafted using the Xformer model and tested on the Restormer model resulted in a classification accuracy of 35.92%, which is comparable to the accuracy of 35.23% when tested on the original Xformer model. This indicates that the adversarial perturbations are not denoising model-specific and can generalize to other architectures.

Moreover, the image quality metrics such as PSNR and SSIM after denoising remain high, suggesting that the transfer does not substantially degrade the visual quality of the images. This is crucial for practical scenarios where the adversarial examples need to remain inconspicuous to human observers.

## D.6    ROBUSTNESS OF ADVERSARIAL EXAMPLES

We assess the robustness of our adversarial examples against several common defense strategies, including Feature Squeezing Xu (2017), Non-Local Means Denoising Buades et al. (2005), and Randomized Smoothing Chiang et al. (2020). As shown in Tab. 15 and Fig. 9, the semantic modifications introduced by our method remain detectable under these defense conditions, with visible alterations in all cases. Notably, these techniques do not significantly degrade image quality, as indicated by stable PSNR and LPIPS values. Moreover, the semantic changes persist despite the application of defenses, as reflected by the high ROUGE-L scores (e.g., 0.80 under Feature Squeezing), which demonstrate the resilience of our adversarial examples to these countermeasures. These results highlight the effectiveness and robustness of our attack against widely used defense strategies.

## D.7    SENSITIVITY OF MIGA TO THE REFERENCE IMAGE

To complement the analysis in Sec. 5.4 on task difficulty, we further study how MIGA depends on the reference image in the unknown–task setting. Figure 10 controls three key factors of the reference:

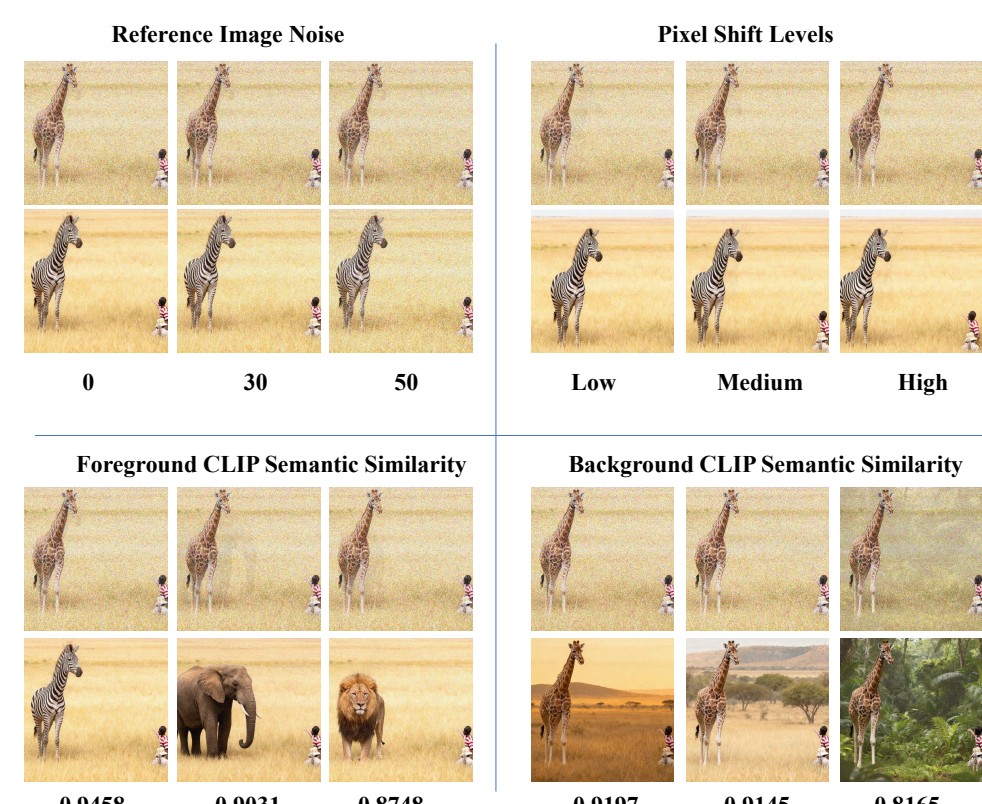

**Figure 10:** Ablation on the reference image in the unknown–task setting. Top-left: we vary the **quality** of the reference image by adding increasing noise (0, 30, 50). Top-right: we vary the **alignment** via low/medium/high pixel shifts. Bottom-left: we change the foreground object (zebra → elephant → lion) while keeping the background fixed and report the foreground CLIP similarity. Bottom-right: we change the background (grassland → savanna → rainforest) while keeping the giraffe fixed and report the background CLIP similarity.

(i) *quality*, by adding different levels of noise/blur to the reference image; (ii) *alignment*, by applying controlled pixel shifts; and (iii) *semantic similarity*, by gradually changing either the foreground object or the background scene. Across moderate degradations of quality and alignment (noise levels 0/30/50 and low/medium pixel shift), the denoised images remain visually similar, indicating that MIGA is robust to imperfect but reasonable references. Moreover, because MIGA is driven by an explicit reference image, the semantic direction and strength of the attack are directly controlled by the chosen reference, rather than arising from an uncontrolled or vague semantic drift. Performance degrades noticeably only when the semantics of the reference become strongly inconsistent with the original image, e.g., replacing the entire grassland background with a rainforest. In this regime, the foreground/background CLIP similarities drop and the attack becomes less effective, which is consistent with the inherent limitation of any reference-guided semantic manipulation: a reference that no longer shares core semantics with the original image cannot reliably guide the desired semantic shift.

## D.8 COMPARISON WITH SEMANTIC ADVERSARIAL EXAMPLES

To further study semantic-aware attacks on denoising models, we compare MIGA with SAE Hosseini & Poovendran (2018) on the ImageNet-10 classification task. Four denoising networks are considered (Xformer, Restormer, PromptIR, AFM), and Table 16 reports both image-quality metrics (PSNR, SSIM, LPIPS, Entropy) and post-denoising classification accuracy.

Table 16: Summary results for adversarial attacks on ImageNet-10. Higher PSNR/SSIM and lower LPIPS/Entropy indicate better visual quality, while lower Accuracy indicates stronger attack success after denoising.

| Network | Image | PSNR↑ | SSIM↑ | LPIPS↓ | Entropy↓ | Accuracy↓ |
|---|---|---|---|---|---|---|
| – | $x$ | $\infty$ | 1.00 | 0.00 | 7.19 | 99.46 |
| | $x_{\mathrm{N}}$ | 17.65 | 0.30 | 0.62 | 7.71 | 83.00 |
| Xformer | $D(x_{\mathrm{N}})$ | 23.63 | 0.65 | 0.41 | 7.04 | 84.73 |
| | $x_{\mathrm{SAE}}$ | 17.19 | 0.27 | 0.67 | 7.76 | 76.48 |
| | $D(x_{\mathrm{SAE}})$ | 17.68 | 0.43 | 0.54 | 7.18 | 37.86 |
| | $x_{\mathrm{MIGA}}$ | 17.62 | 0.30 | 0.63 | 7.72 | 74.69 |
| | $D(x_{\mathrm{MIGA}})$ | 25.64 | 0.67 | 0.22 | 7.13 | 35.23 |
| Restormer | $D(x_{\mathrm{N}})$ | 22.36 | 0.70 | 0.38 | 6.81 | 89.85 |
| | $x_{\mathrm{SAE}}$ | 16.89 | 0.27 | 0.68 | 7.78 | 74.55 |
| | $D(x_{\mathrm{SAE}})$ | 21.55 | 0.65 | 0.48 | 7.13 | 58.24 |
| | $x_{\mathrm{MIGA}}$ | 17.60 | 0.30 | 0.63 | 7.72 | 70.62 |
| | $D(x_{\mathrm{MIGA}})$ | 30.77 | 0.78 | 0.26 | 7.06 | 51.54 |
| PromptIR | $D(x_{\mathrm{N}})$ | 22.83 | 0.50 | 0.45 | 7.56 | 85.77 |
| | $x_{\mathrm{SAE}}$ | 16.64 | 0.26 | 0.68 | 7.76 | 77.26 |
| | $D(x_{\mathrm{SAE}})$ | 15.81 | 0.16 | 0.77 | 7.76 | 37.89 |
| | $x_{\mathrm{MIGA}}$ | 17.61 | 0.30 | 0.63 | 7.72 | 74.46 |
| | $D(x_{\mathrm{MIGA}})$ | 24.20 | 0.59 | 0.37 | 7.60 | 21.31 |
| AFM | $D(x_{\mathrm{N}})$ | 23.01 | 0.71 | 0.42 | 7.01 | 86.43 |
| | $x_{\mathrm{SAE}}$ | 17.11 | 0.28 | 0.63 | 7.80 | 73.44 |
| | $D(x_{\mathrm{SAE}})$ | 16.84 | 0.59 | 0.59 | 6.89 | 58.21 |
| | $x_{\mathrm{MIGA}}$ | 19.32 | 0.32 | 0.62 | 7.63 | 74.92 |
| | $D(x_{\mathrm{MIGA}})$ | 31.28 | 0.78 | 0.21 | 7.03 | 42.99 |

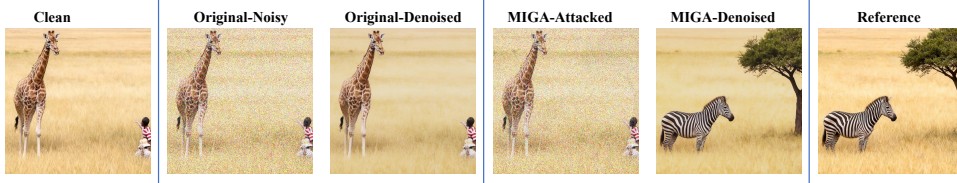

**Clean**    **Original-Noisy**    **Original-Denoised**    **MIGA-Attacked**    **MIGA-Denoised**    **Reference**

Figure 11: Illustration of multi-object replacement under MIGA. From left to right: Clean, Original-Noisy, Original-Denoised, MIGA-Attacked, MIGA-Denoised, and Reference.

Across all denoisers, SAE successfully reduces the accuracy of the downstream classifier, but this effect is accompanied by a noticeable degradation of denoised-image quality: $D(x_{\mathrm{SAE}})$ typically shows lower PSNR/SSIM and higher LPIPS than $D(x_{\mathrm{N}})$.

In contrast, $D(x_{\mathrm{MIGA}})$ consistently achieves higher PSNR/SSIM and lower LPIPS than $D(x_{\mathrm{SAE}})$, while further reducing the classification accuracy. For example, under the AFM denoiser the accuracy drops from 58.21% for $D(x_{\mathrm{SAE}})$ to 42.99% for $D(x_{\mathrm{MIGA}})$, and PSNR increases from 16.84 to 31.28. Similar trends can be observed for Xformer, Restormer, and PromptIR. These results indicate that MIGA provides a more favorable trade-off between semantic manipulation strength and denoised image fidelity than SAE.

### D.9 MULTI-OBJECT REPLACEMENT (MULTI-OBJECT INTERACTIONS)

We further investigate whether MIGA can handle more complex semantic shifts involving multi-object interactions. Figure 11 shows an example where both the main foreground animal and a secondary object in the scene are jointly modified according to a reference image. From left to right, we display the clean image, the noisy input, the denoised result $D(x_{\mathrm{noisy}})$, the MIGA-attacked noisy image $x_{\mathrm{MIGA}}$, the corresponding denoised output $D(x_{\mathrm{MIGA}})$, and the reference image. While $D(x_{\mathrm{noisy}})$ faithfully restores the original giraffe–human scene, $D(x_{\mathrm{MIGA}})$ successfully matches the multi-object semantics of the reference (e.g., replacing the giraffe and the human with the zebra–tree configuration) and remains visually plausible with high image quality. This example indicates that MIGA extends beyond single-object edits and can realize coherent multi-object replacements within a complex scene.

### D.10 ADDITIONAL EVALUATION ON OBJECT DETECTION

To complement the classification experiments in Sec. 5.2, we further evaluate MIGA on a standard object detection downstream task. Concretely, we adopt a DETR-based detector Carion et al. (2020) trained on the COCO dataset and insert a denoising model in front of the detector, following the same denoiser-based pipeline as in the main paper. We consider four representative denoisers (Xformer, Restormer, PromptIR, AFM) and report both image-quality metrics (PSNR, SSIM, LPIPS, entropy) and detection performance (AP).

As summarized in Table 17, MIGA consistently achieves lower AP after denoising than the standard adversarial baseline $x_{\text{adv}}$, indicating a stronger attack on the detector, while simultaneously improving the denoised image quality (higher PSNR and SSIM, lower LPIPS). This trend holds across all four denoisers (Xformer, Restormer, PromptIR, AFM), showing that MIGA is effective not only for classification but also for object detection in denoiser-based pipelines.

### D.11 ADDITIONAL RESULTS WITH VIT CLASSIFIER

To verify that the effectiveness of MIGA is not tied to a specific downstream classifier, we further replace the ResNet classifier in Section 5.2 with a ViT-based classifier and repeat the ImageNet-10 experiments under the same denoiser-based pipeline. The quantitative results are summarized in Table 18.

For the Xformer denoiser, the denoised noisy input $D(x_{\text{noisy}})$ already achieves a high classification accuracy of $82.17\%$ with reasonable image quality (23.57 dB PSNR and 0.40 LPIPS). After applying the standard I-FGSM attack ($x_{\text{adv}}$), the denoised output $D(x_{\text{adv}})$ reduces the ViT accuracy to $42.96\%$, but this comes at a substantial cost in perceptual quality (PSNR drops to 16.03 dB and LPIPS increases to 0.65). In contrast, $D(x_{\text{MIGA}})$ simultaneously improves the denoised image quality and strengthens the semantic attack: it achieves higher PSNR (25.56 dB) and much lower LPIPS (0.20), while further reducing the classifier accuracy to $34.38\%$.

Similar trends are observed for the other denoisers (Restormer, PromptIR, AFM) in Table 18. In all cases, MIGA yields denoised images with equal or better visual quality compared to I-FGSM, while inducing a stronger degradation in ViT classification accuracy. These results confirm that MIGA remains effective under a transformer-based classifier and that its advantage is not restricted to a single network architecture.

### D.12 ADDITIONAL ATTACK BASELINES AND CLASSIFICATION ACCURACY

We further extend the comparison in Section 5.2 by including stronger adversarial baselines—PGD, AutoAttack, and CosPGD—in addition to MIGA. For each denoiser (Xformer, Restormer,

Table 17: Attack performance on the COCO object detection task with a DETR-based detector Carion et al. (2020).

| Network | Image | PSNR↑ | SSIM↑ | LPIPS↓ | Entropy↓ | AP↓ |
|---|---|---|---|---|---|---|
| | $x$ | ∞ | 1.00 | 0.00 | 7.19 | 42.04 |
| | $x_{\text{N}}$ | 17.65 | 0.30 | 0.62 | 7.71 | 29.81 |
| Xformer | $D(x_{\text{N}})$ | 23.50 | 0.64 | 0.42 | 7.02 | 35.26 |
| | $x_{adv}$ | 17.22 | 0.29 | 0.66 | 7.74 | 31.79 |
| | $D(x_{adv})$ | 16.28 | 0.33 | 0.68 | 7.19 | 21.56 |
| | $x_{\text{MIGA}}$ | 17.22 | 0.31 | 0.62 | 7.72 | 32.48 |
| | $D(x_{\text{MIGA}})$ | 25.89 | 0.65 | 0.20 | 7.12 | 19.67 |
| Restormer | $D(x_{\text{N}})$ | 22.52 | 0.70 | 0.39 | 6.83 | 33.01 |
| | $x_{adv}$ | 17.58 | 0.30 | 0.68 | 7.71 | 31.84 |
| | $D(x_{adv})$ | 16.42 | 0.56 | 0.55 | 6.76 | 22.52 |
| | $x_{\text{MIGA}}$ | 17.43 | 0.29 | 0.64 | 7.75 | 32.19 |
| | $D(x_{\text{MIGA}})$ | 30.91 | 0.76 | 0.24 | 7.04 | 20.08 |
| PromptIR | $D(x_{\text{N}})$ | 22.68 | 0.49 | 0.43 | 7.55 | 36.44 |
| | $x_{adv}$ | 16.27 | 0.26 | 0.67 | 7.78 | 31.01 |
| | $D(x_{adv})$ | 12.19 | 0.10 | 0.78 | 7.75 | 24.98 |
| | $x_{\text{MIGA}}$ | 17.63 | 0.30 | 0.61 | 7.75 | 33.77 |
| | $D(x_{\text{MIGA}})$ | 24.00 | 0.59 | 0.36 | 7.56 | 20.07 |
| AFM | $D(x_{\text{N}})$ | 23.42 | 0.73 | 0.41 | 7.04 | 31.26 |
| | $x_{adv}$ | 16.73 | 0.28 | 0.62 | 7.76 | 29.94 |
| | $D(x_{adv})$ | 16.24 | 0.60 | 0.60 | 6.97 | 20.45 |
| | $x_{\text{MIGA}}$ | 19.05 | 0.34 | 0.64 | 7.67 | 29.82 |
| | $D(x_{\text{MIGA}})$ | 31.35 | 0.78 | 0.23 | 7.04 | 17.16 |

Table 18: Summary results for ViT on ImageNet-10.

| Network | Image | PSNR↑ | SSIM↑ | LPIPS↓ | Entropy↓ | Accuracy↓ |
|---|---|---|---|---|---|---|
| | $x$ | ∞ | 1.00 | 0.00 | 7.19 | 99.37 |
| | $x_{\text{N}}$ | 17.65 | 0.30 | 0.62 | 7.71 | 82.69 |
| Xformer | $D(x_{\text{N}})$ | 23.57 | 0.65 | 0.40 | 7.01 | 82.17 |
| | $x_{\text{adv}}$ | 17.02 | 0.30 | 0.66 | 7.76 | 78.31 |
| | $D(x_{\text{adv}})$ | 16.03 | 0.36 | 0.65 | 7.23 | 42.96 |
| | $x_{\text{MIGA}}$ | 17.75 | 0.31 | 0.63 | 7.77 | 72.11 |
| | $D(x_{\text{MIGA}})$ | 25.56 | 0.66 | 0.20 | 7.16 | 34.38 |
| Restormer | $D(x_{\text{N}})$ | 22.49 | 0.72 | 0.40 | 6.77 | 89.10 |
| | $x_{\text{adv}}$ | 17.33 | 0.29 | 0.65 | 7.67 | 76.23 |
| | $D(x_{\text{adv}})$ | 15.97 | 0.58 | 0.58 | 6.71 | 63.81 |
| | $x_{\text{MIGA}}$ | 17.47 | 0.30 | 0.63 | 7.76 | 67.20 |
| | $D(x_{\text{MIGA}})$ | 30.75 | 0.76 | 0.28 | 7.07 | 42.05 |
| PromptIR | $D(x_{\text{N}})$ | 22.92 | 0.48 | 0.45 | 7.60 | 88.24 |
| | $x_{\text{adv}}$ | 16.33 | 0.24 | 0.66 | 7.73 | 62.17 |
| | $D(x_{\text{adv}})$ | 11.83 | 0.12 | 0.82 | 7.78 | 33.61 |
| | $x_{\text{MIGA}}$ | 17.52 | 0.29 | 0.62 | 7.70 | 76.88 |
| | $D(x_{\text{MIGA}})$ | 24.33 | 0.58 | 0.35 | 7.63 | 19.02 |
| AFM | $D(x_{\text{N}})$ | 23.13 | 0.69 | 0.42 | 7.01 | 84.19 |
| | $x_{\text{adv}}$ | 17.29 | 0.26 | 0.62 | 7.73 | 78.01 |
| | $D(x_{\text{adv}})$ | 15.98 | 0.58 | 0.61 | 6.92 | 73.46 |
| | $x_{\text{MIGA}}$ | 19.43 | 0.32 | 0.61 | 7.64 | 73.28 |
| | $D(x_{\text{MIGA}})$ | 31.22 | 0.79 | 0.21 | 6.99 | 37.01 |

PromptIR, AFM) on ImageNet-10, Table 19 reports both the metrics on the attacked images and on their denoised counterparts, including PSNR, SSIM, LPIPS, entropy, and post-denoising classification accuracy.

Focusing on the Xformer case, $D(x_{\text{noisy}})$ attains an accuracy of $84.73\%$ with good image quality ($23.63$ dB PSNR, $0.41$ LPIPS). After denoising PGD, AutoAttack, and CosPGD examples, the classifier accuracies are reduced to $54.82\%$, $49.81\%$, and $48.27\%$, respectively, but the denoised images also exhibit noticeably degraded quality (e.g., PSNR in the range $20.19$–$21.44$ dB and higher LPIPS). In contrast, $D(x_{\text{MIGA}})$ achieves the lowest post-denoising accuracy among all attacks ($35.23\%$), while at the same time providing the best perceptual quality ($25.64$ dB PSNR, $0.67$ SSIM, and $0.22$ LPIPS).

The same pattern holds across Restormer, PromptIR, and AFM in Table 19: MIGA consistently attains the strongest degradation of classification performance after denoising, yet its denoised outputs enjoy equal or superior PSNR/SSIM and lower LPIPS compared to those produced by PGD, AutoAttack, and CosPGD. These extended results give a more comprehensive evaluation and demonstrate that, under denoiser-based pipelines, MIGA provides a more effective semantic attack than standard and advanced adversarial methods, without sacrificing the visual quality of the denoised images.

Table 19: Comparison of different adversarial attacks on ImageNet-10.

| Network | Image | PSNR↑ | SSIM↑ | LPIPS↓ | Entropy↓ | Accuracy↓ |
|---|---|---|---|---|---|---|
| | $x$ | $\infty$ | 1.00 | 0.00 | 7.19 | 99.46 |
| | $x_{\text{N}}$ | 17.65 | 0.30 | 0.62 | 7.71 | 83.00 |
| Xformer | $D(x_{\text{N}})$ | 23.63 | 0.65 | 0.41 | 7.04 | 84.73 |
| | $x_{\text{PGD}}$ | 15.82 | 0.22 | 0.71 | 7.84 | 75.62 |
| | $D(x_{\text{PGD}})$ | 21.44 | 0.46 | 0.54 | 7.04 | 54.82 |
| | $x_{\text{AutoAttack}}$ | 17.81 | 0.28 | 0.65 | 7.79 | 78.59 |
| | $D(x_{\text{AutoAttack}})$ | 21.08 | 0.46 | 0.55 | 7.12 | 49.81 |
| | $x_{\text{CosPGD}}$ | 17.20 | 0.27 | 0.66 | 7.78 | 79.81 |
| | $D(x_{\text{CosPGD}})$ | 20.19 | 0.43 | 0.58 | 7.15 | 48.27 |
| | $x_{\text{MIGA}}$ | 17.62 | 0.30 | 0.63 | 7.72 | 74.69 |
| | $D(x_{\text{MIGA}})$ | 25.64 | 0.67 | 0.22 | 7.13 | 35.23 |
| Restormer | $D(x_{\text{N}})$ | 22.36 | 0.70 | 0.38 | 6.81 | 89.85 |
| | $x_{\text{PGD}}$ | 16.19 | 0.26 | 0.68 | 7.77 | 73.22 |
| | $D(x_{\text{PGD}})$ | 21.91 | 0.71 | 0.42 | 6.87 | 65.82 |
| | $x_{\text{AutoAttack}}$ | 17.31 | 0.29 | 0.63 | 7.62 | 78.10 |
| | $D(x_{\text{AutoAttack}})$ | 20.89 | 0.68 | 0.45 | 6.92 | 63.74 |
| | $x_{\text{CosPGD}}$ | 17.50 | 0.29 | 0.64 | 7.68 | 77.47 |
| | $D(x_{\text{CosPGD}})$ | 20.32 | 0.67 | 0.47 | 6.70 | 61.10 |
| | $x_{\text{MIGA}}$ | 17.60 | 0.30 | 0.63 | 7.72 | 70.62 |
| | $D(x_{\text{MIGA}})$ | 30.77 | 0.78 | 0.26 | 7.06 | 51.54 |
| PromptIR | $D(x_{\text{N}})$ | 22.83 | 0.50 | 0.45 | 7.56 | 85.77 |
| | $x_{\text{PGD}}$ | 15.88 | 0.26 | 0.70 | 7.81 | 65.22 |
| | $D(x_{\text{PGD}})$ | 19.42 | 0.31 | 0.59 | 7.32 | 55.83 |
| | $x_{\text{AutoAttack}}$ | 17.02 | 0.29 | 0.67 | 7.79 | 65.28 |
| | $D(x_{\text{AutoAttack}})$ | 19.75 | 0.34 | 0.53 | 7.60 | 51.98 |
| | $x_{\text{CosPGD}}$ | 16.46 | 0.26 | 0.69 | 7.82 | 64.16 |
| | $D(x_{\text{CosPGD}})$ | 15.17 | 0.18 | 0.74 | 7.69 | 49.66 |
| | $x_{\text{MIGA}}$ | 17.61 | 0.30 | 0.63 | 7.72 | 74.46 |
| | $D(x_{\text{MIGA}})$ | 24.20 | 0.59 | 0.37 | 7.60 | 21.31 |
| AFM | $D(x_{\text{N}})$ | 23.01 | 0.71 | 0.42 | 7.01 | 86.43 |
| | $x_{\text{PGD}}$ | 14.09 | 0.19 | 0.77 | 7.89 | 71.22 |
| | $D(x_{\text{PGD}})$ | 23.49 | 0.70 | 0.39 | 7.08 | 65.89 |
| | $x_{\text{AutoAttack}}$ | 16.89 | 0.25 | 0.66 | 7.73 | 76.24 |
| | $D(x_{\text{AutoAttack}})$ | 20.12 | 0.67 | 0.50 | 6.97 | 64.51 |
| | $x_{\text{CosPGD}}$ | 17.17 | 0.27 | 0.65 | 7.71 | 77.76 |
| | $D(x_{\text{CosPGD}})$ | 19.65 | 0.65 | 0.52 | 7.00 | 63.02 |
| | $x_{\text{MIGA}}$ | 19.32 | 0.32 | 0.62 | 7.63 | 74.92 |
| | $D(x_{\text{MIGA}})$ | 31.28 | 0.78 | 0.21 | 7.03 | 42.99 |

## D.13 MORE EXPERIMENTAL RESULTS VISUALIZATION

To provide a more comprehensive understanding of the effectiveness of our proposed method, we present more visualizations of the attack results on various datasets. Fig. 12 showcases examples from the Tampered-IC13, Synthetic Dataset, MAGICBRUSH, and Style Transfer tasks. These visualizations illustrate how MIGA effectively alters the semantic content of the denoised images in a way that impacts the downstream tasks while keeping the perturbations imperceptible.

In the Tampered-IC13 and Synthetic Dataset examples, the text content in the images is subtly altered post-denoising, leading to significant errors in text recognition tasks. For the MAGICBRUSH and Style Transfer tasks, the stylistic attributes or content of the images are modified, affecting the outcomes of tasks like image synthesis or style transfer. These results demonstrate the versatility and robustness of our method on denoising models with different types of downstream tasks. Furthermore, they underscore the critical need for developing more resilient denoising techniques and robust defenses to counteract such adversarial attacks.

## E THEORETICAL INTERPRETATION OF THE LOSS COMBINATION

Our three loss terms in Eq. 2 are not chosen in an ad-hoc manner, but can be viewed as the Lagrangian relaxation of an ideal constrained mutual-information minimization problem. Recall that our high-level objective is to (i) keep the perturbation imperceptible, (ii) preserve the denoised image quality, and (iii) reduce task-relevant mutual information to induce semantic shifts. Formally, this can be written as the following constrained optimization:

$$\min_{\delta} \quad L_{\text{MI}}(\delta)$$
$$\text{s.t.} \quad L_{\text{con}}(\delta) \leq \varepsilon_{\text{con}}, \quad L_{\text{rec}}(\delta) \leq \varepsilon_{\text{rec}}, \tag{15}$$

where $L_{\text{con}}$ controls the imperceptibility of the perturbation, $L_{\text{rec}}$ constrains the visual quality of the denoised image, and $L_{\text{MI}}$ measures task-relevant mutual information. In the known-task case, Appendix A shows that $L_{\text{MI}}$ is equivalent to our task-relevant mutual information $I(x; D(x_N + \delta) \mid C)$ up to a monotone transformation, so minimizing $L_{\text{MI}}$ is equivalent to minimizing $I(x; D(x_N + \delta) \mid C)$.

The standard Lagrangian of equation 15 is

$$\mathcal{L}(\delta, \lambda_{\text{con}}, \lambda_{\text{rec}}) = L_{\text{MI}}(\delta) + \lambda_{\text{con}}\big(L_{\text{con}}(\delta) - \varepsilon_{\text{con}}\big) + \lambda_{\text{rec}}\big(L_{\text{rec}}(\delta) - \varepsilon_{\text{rec}}\big), \quad (16)$$

with $\lambda_{\text{con}}, \lambda_{\text{rec}} \geq 0$ the Lagrange multipliers. Up to additive constants, this is exactly the weighted loss used in Eq. (2):

$$L_{\text{total}}(\delta) = \lambda_{\text{con}} L_{\text{con}}(\delta) + \lambda_{\text{rec}} L_{\text{rec}}(\delta) + \lambda_{\text{MI}} L_{\text{MI}}(\delta), \quad (17)$$

where $\lambda_{\text{MI}} > 0$ rescales $L_{\text{MI}}$ and the three coefficients jointly play the role of penalty parameters enforcing the constraints in equation 15. Thus, the proposed combination is theoretically grounded as a Lagrangian formulation of "minimize task-relevant mutual information under imperceptibility and reconstruction constraints".

**Relation to multi-objective optimization.** The three losses can also be interpreted as a multi-objective problem with

$$f_1(\delta) = L_{\text{con}}(\delta), \quad f_2(\delta) = L_{\text{rec}}(\delta), \quad f_3(\delta) = L_{\text{MI}}(\delta). \quad (18)$$

We are interested in perturbations $\delta$ that achieve a good trade-off between imperceptibility, denoised image quality, and semantic manipulation. Classical results in multi-objective optimization state that, under mild regularity conditions (continuity and differentiability of the $f_i$), any local minimizer of a strictly positive weighted sum

$$L_{\text{total}}(\delta) = \lambda_1 f_1(\delta) + \lambda_2 f_2(\delta) + \lambda_3 f_3(\delta), \quad \lambda_i > 0, \quad (19)$$

is a Pareto-optimal solution, i.e., no other perturbation can strictly improve all three objectives simultaneously. Conversely, in convex settings, any Pareto-optimal solution can be realized by some choice of weights $\lambda_i$. Therefore, our loss in Eq. (2) implicitly selects a point on the Pareto front of the tri-objective problem, and changing $(\lambda_{\text{con}}, \lambda_{\text{rec}}, \lambda_{\text{MI}})$ moves the solution along the trade-off surface between imperceptibility, denoised quality, and attack strength. This explains why removing or down-weighting any of the three terms in Table 6 leads to degenerate solutions that either sacrifice visual quality or reduce semantic manipulation.

**Gradient-level interaction between the three losses.** Finally, the interaction between the three losses can be made explicit at the gradient level:

$$\nabla_\delta L_{\text{total}} = \lambda_{\text{con}} \nabla_\delta L_{\text{con}} + \lambda_{\text{rec}} \nabla_\delta L_{\text{rec}} + \lambda_{\text{MI}} \nabla_\delta L_{\text{MI}}. \quad (20)$$

The term $\nabla_\delta L_{\text{MI}}$ drives the perturbation towards directions that most effectively reduce task-relevant mutual information (i.e., induce semantic shifts), while $\nabla_\delta L_{\text{con}}$ and $\nabla_\delta L_{\text{rec}}$ act as regularizers that pull the update back into the region of imperceptible perturbations and visually plausible denoised outputs. The weighting coefficients directly control the relative strength of these competing effects, which is consistent with the empirical ablation in Table 6: removing $L_{\text{MI}}$ weakens semantic alteration, while removing $L_{\text{con}}$ or $L_{\text{rec}}$ allows the optimization to drift towards visually degraded but strongly attacked solutions.

# F    LIMITATIONS AND FUTURE WORK.

MIGA is evaluated under controlled conditions with known denoising models and datasets. Its performance in real-world black-box systems or under diverse noise patterns remains to be explored. In future work, we plan to extend MIGA to real-time settings, enhance its transferability across unseen models, and integrate it into broader robustness evaluation pipelines.

# G    REPRODUCIBILITY STATEMENT.

This work highlights a potential vulnerability in denoising-based pipelines, particularly in safety-critical applications. While the proposed method may be misused to design stronger attacks, it also provides a tool for evaluating and improving model robustness. We advocate responsible use for security testing and robustness benchmarking.

# H    THE USE OF LARGE LANGUAGE MODELS (LLMS)

Large language models were used exclusively for language polishing of this manuscript (clarity, grammar, and minor rewording). The models were not involved in study design, data collection, labeling, analysis, or result generation. All edits were author-vetted prior to inclusion. We did not upload proprietary evaluation data beyond the manuscript text, and the authors bear full responsibility for the content. The use of LLMs does not alter the reproducibility or integrity of the reported results.

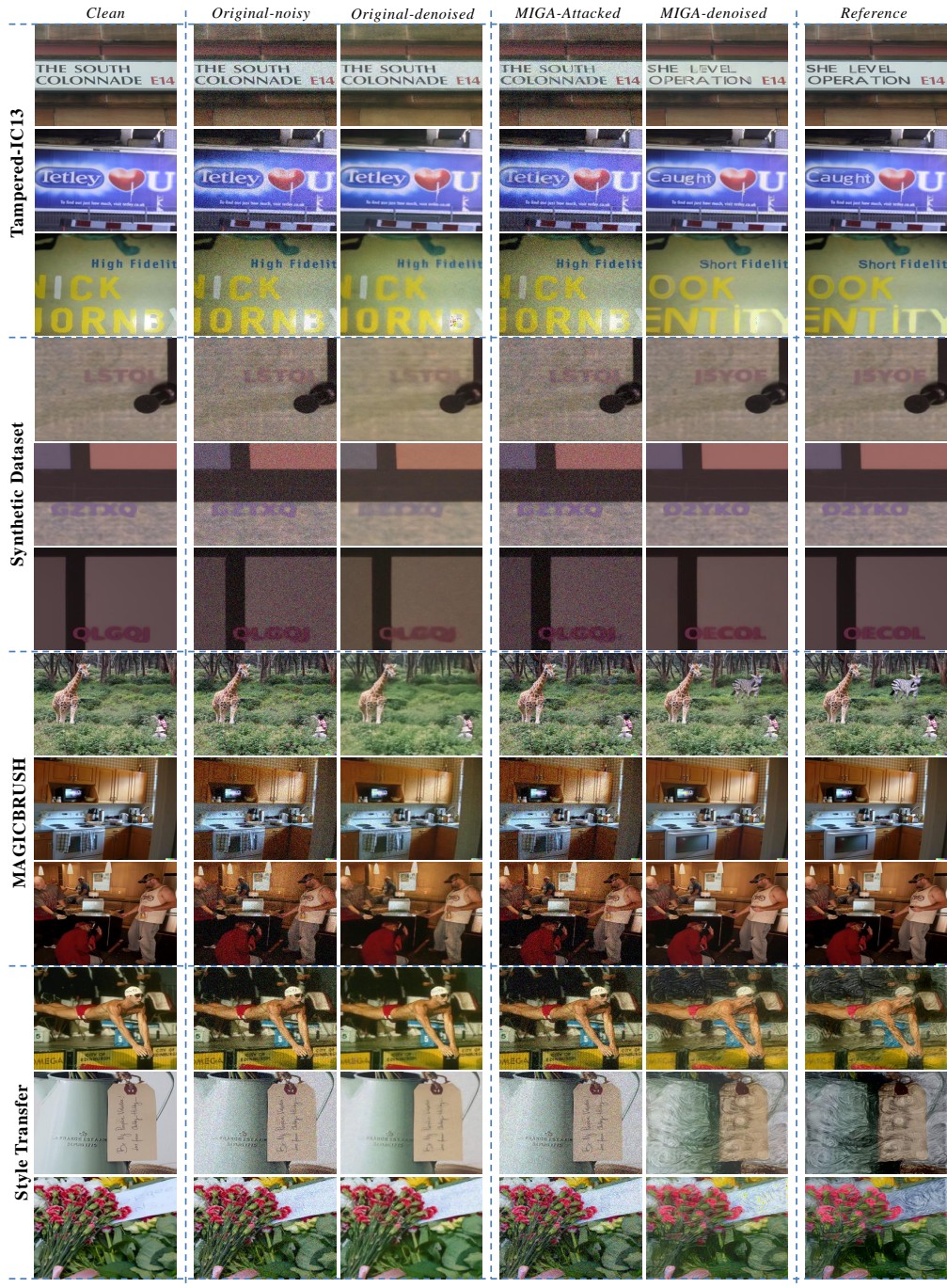

Figure 12: Visualization of MIGA results on Tampered-IC13, Synthetic Dataset, MAGICBRUSH, and Style Transfer.

