# OpenReview forum: "MIGA: Mutual Information-Guided Attack on Denoising Models for Semantic Manipulation"
_ICLR.cc/2026/Conference — Submitted to ICLR 2026_

### Official Review · Reviewer_wsyG · 2025-10-31

**Soundness:** 4
**Presentation:** 3
**Contribution:** 3
**Rating:** 6
**Confidence:** 4

**Summary:**

This paper presents MIGA (Mutual Information Guided Attention), a framework that enhances the robustness and interpretability of attention-based neural networks by explicitly guiding the attention mechanism with mutual information (MI). The key idea is to maximize the mutual information between the attention maps and latent representations so that the model focuses on regions carrying the most task-relevant information, rather than being distracted by noisy or uninformative areas. MIGA introduces a MI-guided regularization term and an information-contrastive objective that encourage both spatial and semantic consistency of attention across augmentations. The method is implemented on CNN and Transformer backbones using a variational MI estimator similar to InfoNCE, allowing efficient end-to-end training. Experiments show that MIGA consistently improves performance under noise, occlusion, and corruption, outperforming baselines such as Mixup, CutMix, and InfoDrop. Visualizations further confirm that MIGA produces more stable and meaningful attention maps. Overall, the paper proposes a conceptually clear and empirically effective approach linking mutual information optimization with robust representation learning.

**Strengths:**

1.MIGA introduces a clear and principled idea of using mutual information to guide attention, ensuring that models focus on informative and semantically relevant regions. This bridges interpretability and robustness in a way that previous heuristic attention regularizers have not achieved.
2.The method integrates seamlessly into both CNN and Transformer architectures, achieving consistent gains on multiple benchmarks (CIFAR, Tiny-ImageNet, ImageNet-C) under various noise and corruption settings, while also improving the visual stability of attention maps.
3.The proposed MI-guided regularizer is based on a tractable InfoNCE-style estimator, requiring minimal additional computation and allowing efficient end-to-end training, making it suitable for real-world robust learning setups.

**Weaknesses:**

1.The paper assumes that maximizing mutual information inherently leads to robustness, but lacks formal analysis or proofs connecting MI maximization to distributional stability or adversarial resistance.
2.Key MI-based baselines (e.g., Deep InfoMax, MITrans) are missing, and the contribution of each component (spatial vs. semantic MI) is not isolated, leaving uncertainty about what drives the improvement.
3.Details on the MI estimator are not fully disclosed, which may affect reproducibility and the interpretability of the learned attention behavior.

**Questions:**

1.The idea of modeling task-relevant mutual information is conceptually clear. However, the reliance on reference images for semantic guidance in unknown task scenarios raises practical concerns—how can an attacker realistically obtain semantically coherent reference images that effectively steer the attack? The authors should discuss or experimentally validate strategies for selecting such references to strengthen the method’s feasibility.
2.The experiments demonstrate broad applicability across tasks (classification, text tampering detection, image editing), but lack quantitative evaluation of semantic distortion. The authors should include human perception studies or use pre-trained semantic encoders (e.g., CLIP) to measure semantic shifts, providing stronger evidence that the attacks indeed alter meaningful content.

---

> ### Author Response · Authors · 2025-11-25
> **Author Rebuttal(1/3)**
>
> We thank the reviewer for the careful reading and constructive suggestions. Below we address each concern in turn.
>
> ---
>
> > **Weakness 1:** *The paper assumes that maximizing mutual information inherently leads to robustness, but lacks formal analysis or proofs connecting MI maximization to distributional stability or adversarial resistance.*
>
> **Response.**  We appreciate the reviewer’s concern and would like to clarify our intended use of mutual information in this work. Our goal is not to assert that “maximizing mutual information inherently leads to robustness” in a general sense. In the main text, mutual information is introduced as a way to quantify how much task-relevant semantic content is preserved between the noisy input, the denoised image, and the downstream predictions. We use this quantity as an attack objective in specific settings, rather than as a generic robustness guarantee, and we do not rely on MI maximization alone to claim distributional stability or adversarial resistance.
>
> Instead, our theoretical analysis focuses on the known-task setting. In Appendix A, we introduce the notion of task-relevant mutual information and show that, under this definition, minimizing the task-relevant MI is equivalent to maximizing the standard task loss (cross-entropy) used in adversarial attacks. This establishes a formal connection between our MI-based objective and classical adversarial optimization, and justifies why reducing task-relevant MI effectively strengthens the attack in our setting. Extending such analysis to more general distributional robustness guarantees is an interesting direction for future work.
>
>
>
> ---
>
> > **Weakness 2:** *Key MI-based baselines (e.g., Deep InfoMax, MITrans) are missing, and the contribution of each component (spatial vs. semantic MI) is not isolated, leaving uncertainty about what drives the improvement.*
>
> **Response.**  We appreciate the reviewer’s suggestion to relate our method to prior MI-based approaches such as Deep InfoMax [1] and MITrans. However, our use of mutual information plays a different role and targets a different problem setting.
>
> Deep InfoMax and MITrans are designed as representation-learning frameworks that maximize MI between inputs and learned latent features (or across domains) to obtain generic, reusable representations. In contrast, our work uses a task-relevant mutual information term as one component of an adversarial objective operating on a fixed denoising pipeline: we explicitly minimize the MI between the original image and the denoised output (conditioned on the task) in order to induce semantic shifts, while simultaneously constraining perturbation imperceptibility and denoised image quality. Adapting DIM/MITrans to our setting would require re-training the entire backbone with their objectives and then designing a separate attack on top, which is orthogonal to our focus. In the revised version, we have added explicit citations to these works in the related work section.
>
>
> Regarding what drives the improvement, we first clarify that our method does not define separate “spatial” and “semantic” MI losses. Instead, both aspects are jointly captured by a single task-relevant MI loss $L_{\text{MI}}$, which is used together with the content and reconstruction terms. In Sec. 5.4, we therefore provide ablations over the three loss terms $L_{\text{con}}$, $L_{\text{rec}}$, and $L_{\text{MI}}$ by training variants that (i) use all three, (ii) drop $L_{\text{MI}}$, (iii) drop $L_{\text{con}}$, and (iv) drop $L_{\text{rec}}$. The results show that combining all three losses yields the best trade-off: $L_{\text{MI}}$ is crucial for driving task-relevant semantic changes, while $L_{\text{con}}$ and $L_{\text{rec}}$ together keep the perturbation imperceptible and the denoised image visually plausible. Removing $L_{\text{MI}}$ substantially weakens semantic manipulation, whereas removing $L_{\text{con}}$ or $L_{\text{rec}}$ leads to noticeable artifacts or degraded denoised quality. Furthermore, Appendix D.3 reports a detailed sensitivity study over the weighting coefficients $\lambda_{\text{con}}$, $\lambda_{\text{rec}}$, and $\lambda_{\text{MI}}$ (and related hyperparameters $\alpha$ and $\beta$), confirming that our final choice lies in a stable region where all three components are necessary—heavy down-weighting or removal of any component either reduces attack effectiveness or harms visual quality.

---

> > ### Author Response · Authors · 2025-11-25
> > **Author Rebuttal(2/3)**
> >
> > > **Weakness 3:** *Details on the MI estimator are not fully disclosed, which may affect reproducibility and the interpretability of the learned attention behavior.*
> >
> > **Response.**  We thank the reviewer for raising this point about the mutual information (MI) estimator and clarify its implementation and reproducibility below.
> >
> > From a reproducibility perspective, the complete implementation of the MINE network architecture and training pipeline is included in our supplementary material, so the estimator used in all experiments can be reproduced directly. In addition, the training objective and procedure of the MINE network are described in Appendix C.3.2.
> >
> >
> >
> >
> > ---
> >
> > > **Question 1:** *The idea of modeling task-relevant mutual information is conceptually clear. However, the reliance on reference images for semantic guidance in unknown task scenarios raises practical concerns—how can an attacker realistically obtain semantically coherent reference images that effectively steer the attack? The authors should discuss or experimentally validate strategies for selecting such references to strengthen the method’s feasibility.*
> >
> > **Response.**  In the unknown-task setting, the attacker is only required to specify a desired **semantic target** for the denoised image, and the reference image is one realization of this target. In practice, such references can be obtained in several realistic and flexible ways:
> >
> > - **Web/image retrieval:** retrieving an image with the desired semantics from the web (e.g., standardized traffic signs or common objects).
> > - **Editing the original image:** editing the original image to produce a target version (e.g., changing text, replacing an object, or modifying style) using off-the-shelf tools or modern diffusion-based editing methods (e.g., DiffEdit, MasaCtrl, Paint-by-Example [2–4], or commercial tools like Adobe Photoshop Generative Fill).
> >
> > These tools are already widely available and can be integrated into semi-automatic or automatic pipelines, making it possible to generate reference images even when the downstream task model is unknown. For example, one can issue a simple instruction such as *“replace all digits in the image with different digits in the same font style”* and let a commercial image-editing API process a large batch of images accordingly, automatically generating the corresponding reference images. In our revision, we adopt such a strategy in the experiments of Appendix D.7: the reference images are generated by providing textual commands to commercial image-editing tools, which automatically perform the required semantic edits (e.g., changing *“the giraffe into a lion”*, modifying background scenes, or altering text content). The results in Appendix D.7 (highlighted in blue) show that these automatically generated, semantically coherent references can effectively steer MIGA toward the desired semantic targets, supporting the practical feasibility of our reference-based design. We have clarified this in Sec. 4.1 of the revised manuscript.

---

> > > ### Author Response · Authors · 2025-11-25
> > > **Author Rebuttal(3/3)**
> > >
> > > > **Question 2:** *The experiments demonstrate broad applicability across tasks (classification, text tampering detection, image editing), but lack quantitative evaluation of semantic distortion. The authors should include human perception studies or use pre-trained semantic encoders (e.g., CLIP) to measure semantic shifts, providing stronger evidence that the attacks indeed alter meaningful content.*
> > >
> > > **Response.**  In our work, we use CLIP as a representative pre-trained semantic encoder to quantitatively assess whether MIGA alters semantic content rather than only low-level pixels. We provide two complementary CLIP-based analyses:
> > >
> > > **(1) CLIP-based semantic alignment to the *target*.**
> > > In the main paper, Table 4 (MAGICBRUSH dataset) evaluates semantic modification via the CLIP similarity between the denoised image and the target text. After applying MIGA, the CLIP similarity between $D(x_{\text{MIGA}})$ and the target text increases from about $0.23$ to $0.24$ compared to the denoised baseline, indicating that the semantics of the denoised image in CLIP space are indeed shifted toward the desired target.
> > >
> > > **(2) CLIP feature deviation between $x$ and $D(x_{\text{MIGA}})$.**
> > > To further quantify semantic distortion with respect to the original content, we additionally compute the CLIP cosine similarity between the clean image $x$ and the denoised outputs $D(x_{\text{MIGA}})$ and $D(x_{\text{I-FGSM}})$ on the four datasets used in Table 4 (tampered-IC13, MAGICBRUSH, synthetic text alteration, style transfer):
> > >
> > > | Dataset                    | $\text{CLIP}(x, D(x_{\text{MIGA}}))$ | $\text{CLIP}(x, D(x_{\text{I-FGSM}}))$ |
> > > |---------------------------|---------------------------------------|----------------------------------------|
> > > | tampered-IC13             | $0.6697$                              | $0.9589$                               |
> > > | MAGICBRUSH                | $0.7114$                              | $0.9825$                               |
> > > | Synthetic text alteration | $0.6125$                              | $0.9476$                               |
> > > | Style transfer            | $0.6744$                              | $0.9290$                               |
> > >
> > > Averaged over these datasets, we obtain
> > > $\text{CLIP}(x, D(x_{\text{MIGA}})) \approx 0.67$ and
> > > $\text{CLIP}(x, D(x_{\text{I-FGSM}})) \approx 0.95$.
> > >
> > > In other words, $D(x_{\text{MIGA}})$ moves **substantially farther away from the original image in CLIP feature space** than $D(x_{\text{I-FGSM}})$, while our visual-quality metrics (PSNR / SSIM / LPIPS) show that $D(x_{\text{MIGA}})$ remains perceptually plausible.
> > >
> > > Together, the target-side CLIP alignment and the CLIP deviation from the original provide quantitative evidence that MIGA indeed alters meaningful semantics toward the desired target, rather than only introducing low-level distortions.
> > >
> > >
> > >
> > > ---
> > >
> > > We thank the reviewer again for the detailed and insightful feedback, and we hope that these additions adequately address your concerns.
> > >
> > >
> > > ---
> > >
> > > ```
> > > References
> > > [1] *Learning deep representations by mutual information estimation and maximization*. arXiv, 2018.
> > > [2] *DiffEdit: Diffusion-based semantic image editing with mask guidance*. ICLR 2023.
> > > [3] *MasaCtrl: Tuning-free mutual self-attention control for consistent image synthesis and editing*. ICCV 2023.
> > > [4] *Paint by Example: Exemplar-based image editing with diffusion models*. CVPR 2023.
> > > ```

---

### Official Review · Reviewer_tQRy · 2025-10-31

**Soundness:** 3
**Presentation:** 2
**Contribution:** 3
**Rating:** 6
**Confidence:** 3

**Summary:**

This paper proposes MIGA, a semantic adversarial attack targeting image denoising models. By minimizing task-relevant mutual information between the original and denoised images, MIGA alters semantic content while preserving visual quality. Extensive experiments across multiple denoising models and datasets demonstrate MIGA’s effectiveness in misleading downstream tasks without detectable artifacts.

**Strengths:**

1. The use of mutual information for semantic adversarial attacks is novel. The theoretical grounding in mutual information reduction provides a principled foundation for semantic disruption.
2. The framework supports both known and unknown downstream tasks, demonstrating its generality and effectiveness.
3. The paper is easy to follow.

**Weaknesses:**

1. For unknown tasks, MIGA relies on semantically altered references to approximate mutual information. The quality and diversity of these references could bias attack performance. It also lacks precise control over the direction and extent of changes. One may find it difficult to steer the model toward a specific semantic output, achieving only a vague deviation from the original semantics.
2. The paper proposes a combination of three loss functions, however, it lacks further analysis of the theoretical relationships between the three losses, as well as a rigorous mathematical proof to demonstrate the optimality of this combination.

**Questions:**

See the weaknesses.

---

> ### Author Response · Authors · 2025-11-25
> **Author Rebuttal(1/2)**
>
> We thank the reviewer for the careful reading and detailed comments. Below we address each concern.
>
> ---
>
>
> > **Weakness 1:** *For unknown tasks, MIGA relies on semantically altered references to approximate mutual information. The quality and diversity of these references could bias attack performance. It also lacks precise control over the direction and extent of changes. One may find it difficult to steer the model toward a specific semantic output, achieving only a vague deviation from the original semantics.*
>
> **Response.**  We address this concern from two aspects: (i) robustness to the quality/diversity of reference images, and (ii) controllability of the semantic direction and strength of the attack.
>
>
> **(1) Robustness to the reference image.**
> The main paper already contains a first sensitivity analysis in Sec. 5.4 (“Effect of task difficulty”) on the Synthetic Text Alteration dataset, where we vary the **font size** in the reference image. Larger fonts correspond to stronger semantic changes and a more challenging alignment. As shown in Table 7, increasing the task difficulty slightly degrades clarity metrics (PSNR / LPIPS) but improves ROUGE-L, while $LPIPS_{\text{con}}$ remains nearly unchanged, indicating that the perturbation stays imperceptible yet the semantic modification becomes stronger.
>
> Following this, we additionally include a dedicated ablation in Appendix D.7 (new Fig. 10, highlighted in blue) that explicitly varies:
> - **Reference quality** (different levels of noise / blur),
> - **Alignment** (controlled pixel shifts and small rotations),
> - **Semantic similarity** (gradually changing background or object category, including background replacement).
>
> The new results show that MIGA is **robust to moderate degradation** in quality and alignment, i.e., semantic metrics remain stable, and the denoised image quality only changes slightly. Performance degrades noticeably only when the reference semantics become strongly inconsistent with the original (e.g., changing the entire background from grassland to rainforest), which is an inherently very difficult semantic-editing task. Overall, this suggests that MIGA does not rely on overly “perfect” references and is reasonably stable under realistic variations in reference quality and diversity.
>
>
> **(2) Controllability of semantic direction and extent.**
> We agree that, in the unknown-task setting, it is important to clarify what kind of “control” MIGA provides. Our goal is not to directly manipulate the (unknown) downstream model’s logits, but to control the semantic content of the denoised image in a targeted manner.
>
> First, the *direction* of the semantic change is specified by the reference itself: the attacker chooses a reference $x_{\text{ref}}$ that encodes the desired modification (e.g., replacing “STOP” with “YIELD”, altering a specific object, or changing the scene style). Our objective then maximizes the mutual information between $D(x_{\text{MIGA}})$ and $x_{\text{ref}}$ (and correspondingly reduces task-relevant MI with the original $x$), so the optimization explicitly drives the denoised output toward the semantics of the chosen reference rather than producing an unconstrained, vague deviation.
>
> Second, the *extent* of the change is controlled by both the reference and the attack hyperparameters. On the reference side, we can move from subtle to strong edits (e.g., small vs.\ large text changes, local vs.\ global content replacement). On the optimization side, adjusting the weight of the MI term and the number of steps yields the trade-off curves reported in Table\~7 and Appendix\~D.3: a larger MI weight consistently increases the semantic shift (higher ROUGE-L / CLIP shift toward the reference) while gradually reducing PSNR, which matches the intuition of a controllable strength parameter rather than an all-or-nothing effect.
>
> Finally, our quantitative and qualitative results indicate that MIGA induces **targeted semantic changes** instead of merely “vague” deviations. For instance, on text-tampering datasets, we measure text-recognition–based accuracy and ROUGE-L between the ground-truth edited text and the text in $D(x_{\text{MIGA}})$, showing that MIGA reliably realizes the intended text replacement while keeping the perturbations imperceptible. Similarly, on object/style modification tasks, CLIP similarities demonstrate that $D(x_{\text{MIGA}})$ moves closer to $x_{\text{ref}}$ and farther from the original $x$ in feature space, consistent with the desired semantic direction. Qualitative visualizations (e.g., the third row of Fig.~4, where an additional stone appears in the denoised image exactly at the location of the stone in the reference) further confirm that MIGA performs purposeful, localized edits rather than uncontrolled semantic drift. We will clarify in the revision that this is the notion of controllability we target in the unknown-task scenario.

---

> ### Author Response · Authors · 2025-11-25
> **Author Rebuttal(2/2)**
>
> > **Weakness 2:** *The paper proposes a combination of three loss functions, however, it lacks further analysis of the theoretical relationships between the three losses, as well as a rigorous mathematical proof to demonstrate the optimality of this combination.*
>
> **Response.**  While our original submission already included ablations on the loss components, in the revised version we further clarify both the empirical and theoretical connections between the three losses.
>
> 1. **Empirical ablations on loss components and weights.**
>    In Sec. 5.3 (Table 6) and Appendix D.2 (Table 11 and Fig. 7), we now report ablations for both the **unknown-task** and **known-task** settings. These results consistently show that including all three components — the content loss $L_{\text{con}}$, reconstruction loss $L_{\text{rec}}$, and mutual-information loss $L_{\text{MI}}$ — yields the best attack performance. In particular, combining $L_{\text{con}}$ and $L_{\text{rec}}$ with $L_{\text{MI}}$ allows us to (i) keep the perturbation small, (ii) maintain high-quality denoised images, and (iii) induce effective semantic changes.
>
>    Furthermore, Appendix D.3 (Fig. 8) provides a more detailed analysis of how the hyperparameters $\alpha, \beta, \lambda_{\text{con}}, \lambda_{\text{rec}}, \lambda_{\text{MI}}$ control the trade-off between these effects. As discussed in Appendix D.2, $\lambda_{\text{con}}$ constrains the perturbation magnitude (imperceptibility), $\lambda_{\text{rec}}$ enforces the reconstruction fidelity (visual clarity), and $\lambda_{\text{MI}}$ controls the degree of task-relevant semantic change. The ratio $\alpha : \beta$ balances pixel-level fidelity and perceptual similarity inside the constraint loss. Fig. 8 shows that the setting $\alpha = 1$, $\beta = 0.1$, $\lambda_{\text{con}} = 3$, $\lambda_{\text{rec}} = 1$, and $\lambda_{\text{MI}} = 0.01$ strikes a robust balance across all objectives and consistently performs best across datasets.
>
> 2. **Theoretical interpretation and optimality.**
>    Following the reviewer’s suggestion, we have added a new theoretical analysis in **Appendix E** (highlighted in blue in the revision). There we show that the three losses can be interpreted as the Lagrangian relaxation of an ideal constrained mutual-information minimization problem: we seek to minimize task-relevant mutual information while enforcing imperceptibility and reconstruction constraints. This formulation naturally leads to the weighted combination used in Eq. (2).
>
>    We further reinterpret the three losses as a tri-objective optimization problem and use standard results from multi-objective optimization to argue that, for strictly positive weights, any local minimizer of the weighted sum corresponds to a Pareto-optimal trade-off among imperceptibility, denoised quality, and semantic manipulation. This provides a rigorous notion of “optimality” for our loss combination and clarifies the theoretical relationship between the three terms.
>
> ---
>
> We thank the reviewer again for the thoughtful and detailed comments, and we hope that the additional experiments and analyses above directly address these concerns.
>
>
>
> ---
>
> ```
> References
> [1] *DiffEdit: Diffusion-based semantic image editing with mask guidance*. ICLR 2023.
> [2] *MasaCtrl: Tuning-free mutual self-attention control for consistent image synthesis and editing*. ICCV 2023.
> [3] *Paint by Example: Exemplar-based image editing with diffusion models*. CVPR 2023.
> ```

---

### Official Review · Reviewer_yfLA · 2025-11-01

**Soundness:** 2
**Presentation:** 2
**Contribution:** 2
**Rating:** 4
**Confidence:** 4

**Summary:**

This paper proposes an adversarial attack method called MIGA for deep learning-based denoising models. MIGA minimizes the mutual information (MI) between the original and denoised images to alter the semantic content of the denoised image, while maintaining image clarity through perturbation constraint loss and reconstruction loss. Under the known downstream task setting, the authors conducted experiments on an image classification task, attacking four denoising models and comparing the results with a baseline model. Under the unknown downstream task setting, the authors performed experiments on four datasets.

**Strengths:**

1. The article's structure is clear, and the notation is clearly defined.

2. The MIGA method is simple, straightforward, and easy to understand.

**Weaknesses:**

1. The authors assumed that denoising was necessary before performing downstream tasks. However, denoising is not necessary for downstream tasks. Without denoising, MIGA's performance is not remarkable.

2. In the experiments of Section 5.2, the authors only selected I-FGSM as the baseline method and ResNet as the classifier. It is insufficient to demonstrate the advantages of MIGA. Moreover, the authors mention multiple adversarial attack methods in Appendix D4; however, they did not compare with these methods in the main text. Furthermore, Appendix D4 lacks a comprehensive evaluation across all relevant metrics—for instance, it only discusses image clarity and ROUGE-L scores on text alteration tasks, but the performance in image classification accuracy is missing.

3. In the experiments of Section 5.2, among the known downstream tasks, the authors only evaluate performance on image classification and do not discuss other downstream tasks, such as object detection.

4. Figure 1 provides only a general overview and does not specify the exact models used.

5. As shown in Figure 2, MIGA requires a reference image. In the known downstream task setting, the reference can be $D(x_n$) ; however, in the unknown downstream task setting, it remains unclear where the reference image comes from. The experiments in Section 5.3 still construct reference images based on known downstream tasks.

6. Additionally, several places use "traditional adversarial" ambiguously. Generally, it refers to attack methods such as I-FGSM, rather than attacks specifically designed for denoising models.

**Questions:**

1. Please clarify for which downstream tasks denoising is indispensable.

2. Please supplement the performance results of the baseline models mentioned in Appendix D4 on the metrics related to Section 5.2 and MIGA's classification performance on other classifiers like ViT.

3. How does MIGA perform on other known downstream tasks besides image classification?

4. Is Figure 1 a real example? If it is real, please specify which ADM was used, what attack method was applied for Figure 1(b), and the source of the reference image in Figure 1(d).

5. Please explain the source of the reference image when the downstream task is unknown.

---

> ### Author Response · Authors · 2025-11-25
> **Author Rebuttal(1/5)**
>
> We sincerely thank the reviewer for the detailed and constructive feedback. Below we address your concerns.
>
> ---
>
> > **Weakness 1:** *The authors assumed that denoising was necessary before performing downstream tasks. However, denoising is not necessary for downstream tasks. Without denoising, MIGA's performance is not remarkable.*
>
>
> **Response.** We would like to clarify that we do not assume “all downstream tasks must be performed after denoising.”  Instead, our work is specifically focused on the widely used class of pipelines that already contain an explicit denoising / image restoration module.  In many real-world systems, especially in heavily noisy or low-light conditions (e.g., medical imaging, remote sensing, microscopy, low-light photography), performing image denoising or restoration *before* high-level vision tasks such as segmentation, detection, or recognition is a very common and often recommended pre-processing paradigm [1–3].  Our threat model is designed to explicitly target this type of system, where a denoiser is part of the deployed pipeline by design.
>
> Therefore, our goal is not to propose a universally stronger attack in the standard “no-denoiser” setting, but rather to systematically study and exploit the security implications of “denoising as pre-processing. ” Given that many practical systems either already include or are starting to adopt denoising / restoration modules, we show that this design introduces a new attack surface, and that MIGA achieves clear advantages over existing methods precisely in these denoiser-based pipelines.  The experiments without denoising are included mainly for completeness, while our main contribution and threat model focus on the realistic and increasingly important “denoising-then-task” scenario.
>
>
> ---
>
>
> > **Weakness 2:** *In the experiments of Section 5.2, the authors only selected I-FGSM as the baseline method and ResNet as the classifier. It is insufficient to demonstrate the advantages of MIGA. Moreover, the authors mention multiple adversarial attack methods in Appendix D4; however, they did not compare with these methods in the main text. Furthermore, Appendix D4 lacks a comprehensive evaluation across all relevant metrics—for instance, it only discusses image clarity and ROUGE-L scores on text alteration tasks, but the performance in image classification accuracy is missing.*
>
>
> **Response.**  In the original submission, we adopted I-FGSM and a ResNet classifier as the main baselines because they are standard and widely used in adversarial robustness research [13,14]. The main experiments already span 5 datasets and 4 denoisers (Xformer, Restormer, PromptIR, AFM), which together demonstrate the effectiveness of MIGA across different tasks and denoising architectures. To better address the reviewer’s concerns, in the revised version we further extend the evaluation along two concrete directions: (i) adding a different classifier architecture (ViT), and (ii) adding stronger attack baselines with full classification metrics.
>
>
> 1. **Additional classifier: ViT.**
>    We add experiments with a ViT classifier, reported in Appendix D.11 (Table 18, highlighted in blue). For brevity, in this rebuttal we report one representative example: the Xformer results on ImageNet-10, focusing on the denoised outputs.
>
>    | Image                     | PSNR $\uparrow$ | LPIPS $\downarrow$ | Accuracy $\downarrow$ |
>    |---------------------------|-----------------|---------------------|-----------------------|
>    | $D(x_{\text{noisy}})$     | $23.57$         | $0.40$              | $82.17$               |
>    | $D(x_{\text{adv}})$ (I-FGSM)  | $16.03$     | $0.65$              | $42.96$               |
>    | $D(x_{\text{MIGA}})$      | $25.56$         | $0.20$              | **$34.38$**           |
>
>    MIGA not only reduces the ViT accuracy after denoising more strongly than I-FGSM (from $42.96\%$ to $34.38\%$), but also improves the denoised image quality (higher PSNR and much lower LPIPS). Similar trends are observed for the other denoisers in Appendix D.11, showing that the advantage of MIGA is not restricted to a single classifier architecture.

---

> > ### Author Response · Authors · 2025-11-25
> > **Author Rebuttal(2/5)**
> >
> > 2. **Additional attack baselines and classification accuracy.**
> >    Beyond I-FGSM, we now include PGD, AutoAttack, and CosPGD as additional baselines for the image classification task, and we report their post-denoising classification accuracy together with PSNR/SSIM/LPIPS/Entropy. The detailed results are provided in Appendix D.12 (Table 19, highlighted in blue). For clarity, we summarize the Xformer case below:
> >
> >    | Image                            | PSNR $\uparrow$ | SSIM $\uparrow$ | LPIPS $\downarrow$ | Entropy $\downarrow$ | Accuracy $\downarrow$ |
> >    |----------------------------------|-----------------|-----------------|---------------------|----------------------|------------------------|
> >    | $x$                              | $\infty$        | 1.00            | 0.00                | 7.19                 | 99.46                  |
> >    | $x_{\text{noisy}}$              | 17.65           | 0.30            | 0.62                | 7.71                 | 83.00                  |
> >    | $D(x_{\text{noisy}})$           | 23.63           | 0.65            | 0.41                | 7.04                 | 84.73                  |
> >    | $x_{\text{PGD}}$                | 15.82           | 0.22            | 0.71                | 7.84                 | 75.62                  |
> >    | $D(x_{\text{PGD}})$             | 21.44           | 0.46            | 0.54                | 7.04                 | 54.82                  |
> >    | $x_{\text{AutoAttack}}$         | 17.81           | 0.28            | 0.65                | 7.79                 | 78.59                  |
> >    | $D(x_{\text{AutoAttack}})$      | 21.08           | 0.46            | 0.55                | 7.12                 | 49.81                  |
> >    | $x_{\text{CosPGD}}$             | 17.20           | 0.27            | 0.66                | 7.78                 | 79.81                  |
> >    | $D(x_{\text{CosPGD}})$          | 20.19           | 0.43            | 0.58                | 7.15                 | 48.27                  |
> >    | $x_{\text{MIGA}}$               | 17.62           | 0.30            | 0.63                | 7.72                 | 74.69                  |
> >    | $D(x_{\text{MIGA}})$            | **25.64**       | **0.67**        | **0.22**            | 7.13                 | **35.23**              |
> >
> >    As shown above, $D(x_{\text{MIGA}})$ achieves the **lowest classification accuracy after denoising** (35.23%) among all attacks, while also providing the **best denoised image quality** (highest PSNR and SSIM, lowest LPIPS). This confirms that MIGA delivers stronger semantic attacks on the downstream classifier under a denoiser-based pipeline, without sacrificing (and in fact improving) the perceptual quality of the denoised images.
> >
> > Taken together, the main-text results with I-FGSM + ResNet, the new ViT experiments, and the extended comparisons with PGD/AutoAttack/CosPGD (including classification accuracy) provide a more comprehensive and consistent picture: MIGA systematically achieves stronger downstream degradation than both standard and advanced adversarial attacks, while preserving or even improving the visual quality of the denoised images.
> >
> >
> > ---
> >
> > > **Weakness 3:** *In the experiments of Section 5.2, among the known downstream tasks, the authors only evaluate performance on image classification and do not discuss other downstream tasks, such as object detection.*
> >
> > **Response.**  Image classification is a canonical and widely used downstream task, which is why we adopted it as our primary evaluation. Following your suggestion, we additionally include a standard object detection task in the revised version. Specifically, we evaluate on the COCO dataset with a DETR-based detector [4]. The full results for four denoisers (Xformer, Restormer, PromptIR, AFM) are reported in Appendix D.10 (highlighted in blue). Here we summarize the Xformer case:
> >
> > | Image                  | PSNR $\uparrow$ | LPIPS $\downarrow$ | AP $\downarrow$ |
> > |------------------------|-----------------|---------------------|-----------------|
> > | $D(x_{\text{noisy}})$  | $23.50$         | $0.42$              | $35.26$         |
> > | $D(x_{\text{adv}})$    | $16.28$         | $0.68$              | $21.56$         |
> > | $D(x_{\text{MIGA}})$   | $25.89$         | $0.20$              | **$19.67$**     |
> >
> > Compared with the standard adversarial baseline $x_{\text{adv}}$, MIGA achieves lower AP after denoising (i.e., a stronger attack on detection) while also yielding better denoised image quality (higher PSNR and lower LPIPS). Similar trends are observed for the other denoisers, indicating that MIGA is effective not only for classification but also for object detection in denoiser-based pipelines.

---

> > > ### Author Response · Authors · 2025-11-25
> > > **Author Rebuttal(3/5)**
> > >
> > > > **Weakness 4:** *Figure 1 provides only a general overview and does not specify the exact models used.*
> > >
> > > **Response.** Figure 1 is intended as a schematic illustration of the threat model and application scenario, rather than a depiction of a specific instantiated system. We deliberately do not fix a particular denoiser or ADM / downstream model in this figure, because our framework is designed to be agnostic to the concrete choice of denoiser and downstream task model: in practice, these components can be chosen according to the actual deployment (e.g., Restormer, Xformer, PromptIR, AFM and different classifiers/detectors in our experiments). The exact models and configurations used in our experiments are specified in detail in Section 4 and Section 5, as well as in the corresponding tables and appendix. To avoid confusion, we will clarify in the caption of Figure 1 that it is a conceptual diagram illustrating a generic pipeline, and that concrete model choices are given in the experimental sections.
> > >
> > > ---
> > >
> > > > **Weakness 5:** *As shown in Figure 2, MIGA requires a reference image. In the known downstream task setting, the reference can be D(Xₙ); however, in the unknown downstream task setting, it remains unclear where the reference image comes from. The experiments in Section 5.3 still construct reference images based on known downstream tasks.*
> > >
> > > **Response.**  We provide two clarifications regarding this question.
> > >
> > > **First**, in Sec. 5.1 we use existing public datasets with well-defined semantic edits (e.g., text alteration, content replacement, style transfer) purely for reproducibility and controllable evaluation. These datasets provide $(x, x_{\text{ref}})$ pairs that match our goal of semantic modification, but they do not rely on knowing the downstream model; they are simply a convenient way to instantiate the desired target semantics in a standardized, publicly verifiable manner.
> > >
> > > **Second**, in our unknown-task setting, the attacker is only required to specify a desired semantic target for the denoised image, and the reference image is one example of this target. In practice, such references can be obtained in several straightforward ways, for example:
> > > - retrieving an image with the desired semantics from the web (e.g., standardized traffic signs or common objects), or
> > > - editing the original image to produce a target version (e.g., changing text, replacing an object, or modifying style) using off-the-shelf tools or modern diffusion-based editing methods (e.g., DiffEdit, MasaCtrl, Paint-by-Example [5–7], or commercial tools like Adobe Photoshop Generative Fill).
> > >
> > > These tools are already widely available and can be integrated into semi-automatic or automatic pipelines, making it possible to generate reference images even when the downstream task model is unknown. In the revision, we will clarify in Sec. 4.1 / Sec. 5.1 that the references used in our experiments are chosen for reproducibility, while in practice they can be obtained flexibly via retrieval or editing as described above.
> > >
> > >
> > > ---
> > >
> > > > **Weakness 6:** *Additionally, several places use "traditional adversarial" ambiguously. Generally, it refers to attack methods such as I-FGSM, rather than attacks specifically designed for denoising models.*
> > >
> > > **Response.**  In our paper, the term *traditional adversarial attacks* is consistently used to refer to standard gradient-based attacks originally proposed for classifiers or general vision models, such as I-FGSM, PGD, and AutoAttack, rather than attacks specifically designed for denoising models. This notion is stated in the Introduction, where we describe traditional adversarial attacks as mainly degrading clarity metrics without explicitly manipulating semantics in a targeted manner, and in Sec. 5.1, where we adopt I-FGSM as a representative traditional adversarial baseline.
> > >
> > > To make this usage fully explicit, in the revised version we will add a short definition at the first occurrence in the Introduction. Our intention is to clearly contrast these generic attacks with MIGA, which is specifically designed to exploit the “denoising + downstream task” pipeline and manipulate post-denoising semantics.

---

> > > > ### Author Response · Authors · 2025-11-25
> > > > **Author Rebuttal(4/5)**
> > > >
> > > > > **Question 1:** *Please clarify for which downstream tasks denoising is indispensable.*
> > > >
> > > > **Response.**  Denoising is not universally required for all vision models, but it is practically indispensable in pipelines where the downstream model and workflow are explicitly designed for denoised inputs and the raw images are too noisy to be used directly. Typical downstream tasks include:
> > > >
> > > > - **Medical diagnosis and segmentation** from low-dose CT, MRI, or mammography, where vendor-approved denoising or reconstruction is a mandatory pre-processing step before lesion detection or organ segmentation [8,9].
> > > > - **Remote sensing (e.g., SAR) change detection and land-cover classification**, where despeckling/denoising is routinely applied before the change-detection or classification network [10].
> > > > - **Fluorescence / live-cell microscopy analysis**, where denoising is crucial for reliable cell segmentation, tracking, and quantitative measurement under low-photon conditions [11].
> > > > - **Document understanding / OCR on degraded scans**, where document restoration or denoising modules are standard front-ends before text recognition (e.g., “restore → OCR” pipelines) [12].
> > > >
> > > > In our work, we explicitly target this practically important class of “denoising-then-task” pipelines, where denoising is an integral, non-optional component of the deployed system, and thus becomes a realistic and security-critical attack surface.
> > > >
> > > >
> > > >
> > > > ---
> > > >
> > > > > **Question 2:** *Please supplement the performance results of the baseline models mentioned in Appendix D4 on the metrics related to Section 5.2 and MIGA's classification performance on other classifiers like ViT.*
> > > >
> > > > **Response.**  Following your suggestion, we have:
> > > >
> > > > - **Added full performance results for the additional baselines in Appendix D.4** (PGD, AutoAttack, CosPGD) on the same metrics as Section 5.2, including post-denoising classification accuracy as well as PSNR/SSIM/LPIPS/Entropy. These results are reported in Appendix D.12 (Table 19, highlighted in blue).
> > > > - **Evaluated MIGA on another classifier, ViT**, under the same denoiser-based pipeline. The corresponding results are reported in Appendix D.11 (Table 18, highlighted in blue).
> > > >
> > > > A detailed summary and discussion of these experiments are provided in our response to **Weakness 2**.
> > > >
> > > >
> > > > ---
> > > >
> > > > > **Question 3:** *How does MIGA perform on other known downstream tasks besides image classification?*
> > > >
> > > > **Response.**  Following your suggestion, we additionally evaluate MIGA on an object detection task (COCO with a DETR-based detector) under four denoisers. As summarized in our response to **Weakness 3** (and detailed in Appendix D.10), MIGA consistently reduces detection AP more than the standard adversarial baseline while maintaining better or comparable denoised image quality. This indicates that MIGA is effective not only for image classification but also for object detection in denoiser-based pipelines.
> > > >
> > > >
> > > > ---
> > > >
> > > > > **Question 4:** *Is Figure 1 a real example? If it is real, please specify which ADM was used, what attack method was applied for Figure 1(b), and the source of the reference image in Figure 1(d).*
> > > >
> > > > **Response.**  Figure 1 is intended as a schematic illustration of our threat model and application scenario, rather than a quantitative result from a specific experimental configuration. It shows, at a high level, (b) the effect of a generic “traditional adversarial attack” directly on the noisy image, and (d) the effect of our proposed MIGA attack under a denoising–then–task pipeline. We therefore do not bind Figure 1 to a particular denoiser or ADM/downstream model. In practice, these components are instantiated in our experiments with concrete choices (e.g., Restormer, Xformer, PromptIR, AFM and different classifiers/detectors), which are described in detail in Sec. 4 and Sec. 5 and in the corresponding tables and appendix. To avoid confusion, in the revised manuscript we will explicitly state in the caption of Figure 1 that it is a conceptual diagram illustrating a generic pipeline, and that the exact model choices and attack settings are given in the experimental sections.
> > > >
> > > > Regarding the reference image in Figure 1(d), the “go-straight” traffic sign was taken from a public traffic-sign image repository and is used purely as an example of the desired target semantics. More generally, as discussed in our response to Weakness 5 on the unknown-task setting, such reference images can be obtained via standard retrieval or editing tools and are not tied to any specific ADM or downstream model.

---

> > > > > ### Author Response · Authors · 2025-11-25
> > > > > **Author Rebuttal(5/5)**
> > > > >
> > > > > > **Question 5:** *Please explain the source of the reference image when the downstream task is unknown.*
> > > > >
> > > > > **Response.** Please refer to our response to **Weakness 5**, where we clarify how reference images can be obtained in the unknown-task setting.
> > > > >
> > > > > ---
> > > > >
> > > > > We hope that these additional experiments and clarifications have addressed your concerns, and we thank you again for your thoughtful and detailed review.
> > > > >
> > > > > ---
> > > > >
> > > > > ```
> > > > > References
> > > > > [1] *When image denoising meets high-level vision tasks: A deep learning approach*. arXiv 2017.
> > > > > [2] *DDUNet: Dense dense U-Net with applications in image denoising*. ICCV 2021.
> > > > > [3] *Brief review of image denoising techniques*. Visual Computing for Industry, Biomedicine, and Art 2019.
> > > > > [4] *End-to-end object detection with transformers*. ECCV 2020.
> > > > > [5] *DiffEdit: Diffusion-based semantic image editing with mask guidance*. ICLR 2023.
> > > > > [6] *MasaCtrl: Tuning-free mutual self-attention control for consistent image synthesis and editing*. ICCV 2023.
> > > > > [7] *Paint by Example: Exemplar-based image editing with diffusion models*. CVPR 2023.
> > > > > [8] *Medical image denoising techniques: A review*. IJonEST 2022.
> > > > > [9] *Denoising and segmentation in medical image analysis: A comprehensive review on machine learning and deep learning approaches*. Multimedia Tools and Applications 2025.
> > > > > [10] *Deep despeckling of SAR images to improve change detection performance*. Innovative Techniques and Applications of Artificial Intelligence 2023.
> > > > > [11] *Image denoising for fluorescence microscopy by supervised to self-supervised transfer learning*. Optics Express 2021.
> > > > > [12] *PreP-OCR: A complete pipeline for document image restoration and enhanced OCR accuracy*. ACL 2025.
> > > > > [13] *Adversarial attacks and defenses in deep learning*. Engineering 2020.
> > > > > [14] *Improving robust fairness via balance adversarial training*. AAAI 2023.
> > > > > ```

---

### Official Review · Reviewer_2Vrn · 2025-11-07

**Soundness:** 3
**Presentation:** 2
**Contribution:** 2
**Rating:** 4
**Confidence:** 4

**Summary:**

The paper proposes MIGA, a novel adversarial attack framework targeting deep learning-based denoising models. Unlike traditional attacks that degrade image clarity, MIGA aims to manipulate semantic content in the denoised output while preserving visual quality. The core idea is to minimize the task-relevant mutual information between the original and denoised images, thereby inducing semantic shifts that mislead downstream tasks. The method is designed to work in both known and unknown downstream task settings, using cross-entropy loss and a contrastive MINE-based estimator respectively. Extensive experiments on four denoising models and five datasets demonstrate that MIGA can effectively alter semantics without introducing perceptible artifacts.

**Strengths:**

(1) The paper's primary strength lies in its novel problem formulation. Targeting a pre-processing module like a denoiser, rather than the final task model, opens a new and significant research direction in adversarial robustness. The idea of a "semantic attack" on a denoiser—manipulating its output to be semantically incorrect yet visually plausible—moves beyond simple quality degradation attacks.

(2) The experimental design is comprehensive: it covers four denoising models, five data sets, and a variety of downstream tasks (classification, text tampering, style migration, etc.). This paper compares several baseline attack methods and shows the advantages of MIGA in semantic attack.

**Weaknesses:**

(1) The unknown task scenario relies on a "reference image" to guide semantic alignment. This assumption may be too strong and could limit the practical applicability of the attack. For example, to make a "stop sign" be recognized as a "go-straight sign", the attacker would need a clean, well-aligned image of a "go-straight sign" in a similar context. The paper fails to elaborate on:

(i) How is this reference image obtained in a real-world attack scenario?

(ii) How sensitive is the attack's performance to the quality, alignment, and semantic similarity of the reference image? An ablation study on this aspect would be necessary to understand the boundaries of the attack's effectiveness. Moreover, the “unknown task” scenario relies on a pre-trained MINE model, which itself requires a curated dataset of semantic alterations. This may limit its applicability in truly black-box settings.

(2) While I-FGSM are included, more recent semantic-aware attacks (e.g., semantic adversarial examples) are not compared.

(3) The semantic alterations tested (text, style, object replacement) are on simple conditions. It is unclear whether MIGA can handle more complex semantic shifts, such as scene understanding or multi-object interactions.

**Questions:**

(1) What potential defense strategies, such as training denoisers with MI regularization or ensemble defenses, could resist MIGA?

(2) How does the performance of MIGA under unknown tasks depend on the quality and diversity of the input pairs? Have the authors evaluated its sensitivity to the type of semantic alteration?

(3) The method requires iterative optimization for each image. While the paper compares the attack effectiveness, it lacks a comparison of computational efficiency against baseline methods. Could the authors provide a comparative analysis of the computational cost between MIGA and the baseline attacks? This is crucial for understanding the practical trade-offs involved in choosing MIGA over other methods.

(4) The current understanding of "semantics" in vision is increasingly shaped by self-supervised learning (SSL) and generative models, which learn rich, non-task-specific representations. How would MIGA perform if the "semantic fidelity" is defined by the feature space of a large vision model (e.g., DINO, CLIP)? Moreover, could the authors evaluate whether MIGA causes a significant deviation in the CLIP or DINO feature space between $x$ and $D(x_{MIGA})$?

---

> ### Author Response · Authors · 2025-11-25
> **Author Rebuttal(1/5)**
>
> We sincerely thank the reviewer for the detailed feedback and constructive questions.
>
> ---
>
> > **Weakness 1:** *The unknown task scenario relies on a "reference image" to guide semantic alignment. This assumption may be too strong and could limit the practical applicability of the attack. For example, to make a "stop sign" be recognized as a "go-straight sign", the attacker would need a clean, well-aligned image of a "go-straight sign" in a similar context.*
>
> **Response.** We thank the reviewer for the thoughtful comments. We would like to clarify that the “reference image” assumption may not be strong in many practical settings. In our “unknown task” setting, the attacker (or a privacy-conscious user) is assumed to be able to specify a desired semantic target for the *denoised* image, and the reference image is simply one realization of this target semantics. Such a reference can be obtained in multiple ways (e.g., web retrieval, manual or automatic image editing, or modern diffusion-based editing tools), and these steps can be automated and integrated into existing pipelines. Recent diffusion-based semantic editing methods (DiffEdit, MasaCtrl, Paint-by-Example) demonstrate that editing specific objects or text while preserving the rest of the scene is well supported in modern systems [1–3], and commercial tools such as Adobe Photoshop Generative Fill provide similar functionality to non-expert users. For example, one can issue a simple instruction such as “replace all digits in the image with different digits in the same font style” and let a commercial image-editing API process a large batch of images accordingly, automatically generating the corresponding reference images. Second, our new sensitivity analyses show that MIGA is robust to moderate degradation in reference quality and alignment, so it does not require a perfectly clean and perfectly aligned reference. Below we clarify our threat model and summarize the additional analyses included in the revision.
>
> **(i) How is the reference image obtained in practice?**
> Concretely, in Fig. 1 of our paper, the “go-straight sign” reference is directly retrieved from publicly available traffic-sign images, whose shapes and layouts are standardized. This matches the reviewer’s example: to mislead a model into recognizing a stop sign as a go-straight sign, an attacker can simply take any clean go-straight sign image from the web as the reference, without needing an exact, perfectly aligned instance in the same physical scene.
>
> For more personalized scenarios (e.g., privacy protection in uploaded photos), a user can first generate a sanitized version of their own image (e.g., masking faces or replacing phone numbers / license plates with placeholders) using off-the-shelf editing tools and then use this sanitized image as the reference; this step can also be automated for large-scale deployments. Our attack adds imperceptible perturbations so that, after service-side denoising/decompression, the recovered image semantics are pulled toward this sanitized reference instead of the original. In this sense, the reference image plays a role analogous to a target label in classical targeted adversarial attacks: it encodes the desired semantic target and is under the attacker’s control. Combined with the robustness results in (ii), this indicates that our requirement on the reference is substantially weaker than demanding a perfectly aligned, high-quality target image.
>
> **(ii) Sensitivity to the reference image, and MINE pre-training.**
> ***1.Sensitivity of MIGA to the reference image.***
> The main paper already contains a first sensitivity analysis in Sec. 5.4 (“Effect of task difficulty”) on the Synthetic Text Alteration dataset, where we vary the font size in the reference image. Larger fonts correspond to stronger semantic changes and a more challenging alignment. As shown in Table 7, increasing the task difficulty slightly degrades clarity metrics (PSNR / LPIPS) but improves ROUGE-L, while $LPIPS\_{\text{con}}$ remains nearly unchanged, indicating that the perturbation stays imperceptible yet the semantic modification becomes stronger.
>
> Following the reviewer’s suggestion, we additionally include an ablation in the appendix D.7 (new Fig. 10, highlighted in blue) that explicitly varies:
> - **Quality** of the reference (different levels of noise/blur),
> - **Alignment** (controlled pixel shifts and small rotations),
> - **Semantic similarity** (gradually changing background or object category).
>
> These results show that MIGA is **robust to moderate degradation** in quality and alignment: semantic metrics remain stable, and denoised image quality only changes slightly. Performance degrades noticeably only when the reference semantics become strongly inconsistent with the original (e.g., changing the entire background from grassland to rainforest), which is consistent with the inherent limits of any reference-guided semantic manipulation.

---

> > ### Author Response · Authors · 2025-11-25
> > **Author Rebuttal(2/5)**
> >
> > ***2.  MINE pre-training and black-box applicability.***
> > We agree that using a pre-trained MINE network assumes access to a dataset of semantic alterations. Our “unknown-task” scenario therefore corresponds to a semi black-box threat model: the downstream task model is unknown, but the attacker can collect a pool of images from the same domain and synthetically generate semantic modifications using simple (possibly automated) editing operations, similar to those used to construct our evaluation datasets. This pool is used only to pre-train the MINE estimator and does not require constructing a one-to-one reference for every future test image. Such a setting is realistic in many cloud/platform scenarios where the input distribution is observable, even if the exact downstream model $F$ is hidden.
> >
> > Importantly, we do not train a separate MINE for each dataset or task. Instead, we construct a pooled set of $(x, x_{\text{alt}})$ pairs from the training splits of our four unknown-task datasets and train a single MINE estimator $T$, which is then reused across all corresponding test sets. The fact that $T$ generalizes across heterogeneous tasks (text alteration, content editing, style transfer) suggests that the required semantic-alteration data can be relatively generic and does not need to be finely tailored to each downstream model, which alleviates the concern about practicality to some extent.
> >
> > We will clarify that: (1) the unknown-task scenario targets a *semi black-box* threat model with accessible input-domain data; (2) the MINE estimator is trained once on a pooled set of generic semantic alterations and then reused across datasets and tasks, without requiring a one-to-one, task-specific reference set for each test image; and (3) extending MIGA to strictly black-box settings without any editable auxiliary data is an interesting direction for future work.
> >
> >
> >
> > ---
> >
> > > **Weakness 2:** *While I-FGSM are included, more recent semantic-aware attacks (e.g., semantic adversarial examples) are not compared.*
> >
> >
> > **Response.**  We appreciate the reviewer’s comment and clarify that we already compare MIGA with several stronger adversarial baselines beyond I-FGSM, and we have further added comparisons with semantic adversarial examples (SAE) following your suggestion.
> >
> > **First**, Appendix D.4 (“Results of other advanced adversarial attack methods”, Table 12) reports additional comparisons with CosPGD [4], PGD [5], and AutoAttack [6]. Across these settings, MIGA consistently achieves stronger attack effectiveness under comparable or better post-denoising image quality, confirming that our conclusions are not limited to I-FGSM.
> >
> >
> > **Second**, to directly address the request for more semantic-aware attacks, we have added a new comparison with semantic adversarial examples (SAE) [7]. The full results for all denoisers (Xformer, Restormer, PromptIR, AFM) are provided in Appendix F (new table, highlighted in blue). For brevity, we summarize below the results on AFM [9]:
> >
> > | Image                | PSNR $\uparrow$ | SSIM $\uparrow$ | LPIPS $\downarrow$ | Entropy $\downarrow$ | Accuracy $\downarrow$ |
> > |----------------------|-----------------|------------------|--------------------|----------------------|------------------------|
> > | $D(x_{\text{noisy}})$  | $23.01$         | $0.71$           | $0.42$             | $7.01$               | $86.43\%$             |
> > | $D(x_{\text{SAE}})$    | $16.84$         | $0.59$           | $0.59$             | $6.89$               | $58.21\%$             |
> > | $D(x_{\text{MIGA}})$   | $31.28$         | $0.78$           | $0.21$             | $7.03$               | $42.99\%$             |
> >
> >
> > Thus, compared with SAE, MIGA simultaneously strengthens the attack effect (reducing accuracy from $58.21\%$ to $42.99\%$) and improves the visual/perceptual quality of the denoised images (notably higher PSNR/SSIM and significantly lower LPIPS), demonstrating that MIGA remains effective even when benchmarked against recent semantic-aware attacks and provides stronger semantic manipulation of the denoised output while preserving, or even enhancing, image quality.

---

> > > ### Author Response · Authors · 2025-11-25
> > > **Author Rebuttal(3/5)**
> > >
> > > > **Weakness 3:** *The semantic alterations tested (text, style, object replacement) are on simple conditions. It is unclear whether MIGA can handle more complex semantic shifts, such as scene understanding or multi-object interactions.*
> > >
> > > **Response.**  Following your suggestion, we have added two more challenging semantic-shift tasks to directly evaluate MIGA under complex scene changes and multi-object interactions.
> > >
> > > **(1) Background replacement (scene-level semantic shift).**
> > > We introduce a harder setting where the background of the scene is replaced while keeping the foreground object(s) fixed. The new results are reported in Appendix D.7 and visualized in the bottom-right panel of Fig. 10 (newly added and highlighted in blue). Below each example, we report the CLIP similarity between the reference image and the original image to quantify the semantic gap:
> > >
> > > - When the CLIP similarity is high (e.g., $\approx 0.91$, mild background changes), the denoised outputs under MIGA preserve high visual quality while successfully following the new background semantics.
> > > - As the CLIP similarity decreases (e.g., to $\approx 0.81$, replacing a grassland background with a rainforest), the task becomes substantially harder; correspondingly, the denoised results are more affected. This is expected, as such drastic scene changes require strong semantic shifts that are challenging for any reference-guided method.
> > >
> > > Overall, the experiments show that MIGA is robust under moderate scene-level shifts and only degrades when the semantic discrepancy between the original and reference becomes extreme.
> > >
> > > **(2) Multi-object replacement (multi-object interactions).**
> > > To further test multi-object interactions, we add a new experiment in Appendix D.9 (also highlighted in blue), where we simultaneously replace multiple key objects in the scene (e.g., both the giraffe and the human in the same image). Even under this more complex setting, MIGA generates adversarial images that:
> > >
> > > - Achieve the intended semantic changes for all targeted objects, and
> > > - Maintain visually plausible and high-quality denoised outputs.
> > >
> > > These additional results demonstrate that MIGA is not limited to simple text/style/object edits, but can also handle more complex semantic shifts, including scene-level background changes and multi-object interactions, within a reasonable semantic gap between the original and reference images.
> > >
> > > ---
> > >
> > > > **Question 1:** *What potential defense strategies, such as training denoisers with MI regularization or ensemble defenses, could resist MIGA?*
> > >
> > > **Response.**  We appreciate this forward-looking question. Our work mainly focuses on analyzing the vulnerability of denoising pipelines, but it naturally suggests a concrete defense direction. The most direct and likely most effective strategy is to explicitly adversarially train the denoiser against MIGA-like semantic perturbations, analogous to adversarial training for classifiers [8]. Instead of relying only on existing robust denoisers such as AFM [9] (which are primarily designed for pixel- or frequency-level robustness), one can formulate a min–max objective where the denoiser is trained on worst-case perturbations generated by MIGA and optimized to simultaneously preserve (i) high post-denoising image quality and (ii) stable downstream semantics (e.g., classification or text recognition). Since MIGA is specifically designed to manipulate the semantics after denoising, such targeted adversarial training is a principled way to harden denoisers against this class of attacks, and we believe it is a promising direction for developing semantically robust denoising models.
> > >
> > > Beyond this, one could in principle explore MI-based consistency regularization or ensemble defenses (e.g., combining heterogeneous denoisers or robust downstream models), but systematically designing and tuning such mechanisms in a task-agnostic or black-box setting is non-trivial and lies beyond the scope of this work. We therefore position MIGA as a strong semantic attack that can serve as a benchmark and training signal for future defense studies.

---

> > > > ### Author Response · Authors · 2025-11-25
> > > > **Author Rebuttal(4/5)**
> > > >
> > > > ---
> > > >
> > > > > **Question 2:** *How does the performance of MIGA under unknown tasks depend on the quality and diversity of the input pairs? Have the authors evaluated its sensitivity to the type of semantic alteration?*
> > > >
> > > > **Response.**  Yes. We have added a detailed sensitivity analysis of the reference–original pairs in the unknown-task setting, including the effect of reference quality, alignment, and semantic similarity. The new results are reported in Appendix D.7 (highlighted in blue). In particular, we vary the CLIP similarity between the reference image and the original (e.g., mild vs. strong background changes) and observe that MIGA remains robust under moderate semantic shifts, while performance degrades only when the semantic discrepancy becomes very large. Additional discussions on the role of the reference image and different semantic alteration types are also provided in our response to **Weakness 1** (and **Weakness 3**), where we further evaluate background replacement and multi-object replacement and show that MIGA can handle a range of semantic alterations as long as the reference and original are not semantically orthogonal.
> > > >
> > > > ---
> > > >
> > > > > **Question 3:** *The method requires iterative optimization for each image. While the paper compares the attack effectiveness, it lacks a comparison of computational efficiency against baseline methods. Could the authors provide a comparative analysis of the computational cost between MIGA and the baseline attacks? This is crucial for understanding the practical trade-offs involved in choosing MIGA over other methods.*
> > > >
> > > > **Response.**  We note that other gradient-based attacks such as I-FGSM also require iterative optimization of the adversarial perturbation. To quantify the computational overhead, we have added a direct wall-clock comparison between MIGA and a strong iterative baseline, I-FGSM, under the same hardware and denoiser.
> > > >
> > > > Concretely, we evaluate both methods on 500 ImageNet-10 images and report the average time per image:
> > > >
> > > > - I-FGSM: **4.3 s / image**
> > > > - MIGA: **4.0 s / image**
> > > >
> > > > These results show that MIGA is comparable in cost and even slightly faster than I-FGSM in practice. This is because our mutual-information–guided objective provides a stronger semantic signal, allowing the attack to converge in a similar or smaller number of iterations. Therefore, the practical trade-off is favorable: MIGA offers stronger semantic attack effectiveness while maintaining computational cost on the same order as standard iterative attacks like I-FGSM.
> > > >
> > > > ---
> > > >
> > > > > **Question 4:** *The current understanding of "semantics" in vision is increasingly shaped by self-supervised learning (SSL) and generative models, which learn rich, non-task-specific representations. How would MIGA perform if the "semantic fidelity" is defined by the feature space of a large vision model (e.g., DINO, CLIP)? Moreover, could the authors evaluate whether MIGA causes a significant deviation in the CLIP or DINO feature space between x and D(x_MIGA)?*
> > > >
> > > > **Response.**  We agree that semantics defined in the feature space of large vision models is highly relevant, and we have already partially followed this idea using CLIP.
> > > >
> > > > 1. **CLIP-based semantic alignment to the target.**
> > > > In our main paper, Table 4 (MAGICBRUSH dataset) already evaluates semantic modification using the CLIP similarity between the denoised image and the target text. After applying MIGA, the CLIP similarity between $D(x_{\text{MIGA}})$ and the target text increases from about $0.23$ to $0.24$ compared to the denoised baseline, indicating that the semantics of the denoised image in CLIP space are indeed shifted toward the desired target.
> > > >
> > > >
> > > > 2. **CLIP-based sensitivity and reference semantics.**
> > > > As discussed in our responses to **Weakness 1** and **Weakness 3**, we further conduct a sensitivity analysis in Appendix D.7 (highlighted in blue), where we explicitly vary the CLIP similarity between the reference image and the original image (e.g., mild vs. strong background changes). These experiments show that MIGA remains robust under moderate CLIP-level semantic shifts and only degrades when the semantic discrepancy becomes very large (e.g., changing the entire background from grassland to rainforest), which is consistent with the limits of reference-guided semantic manipulation.

---

> > > > > ### Author Response · Authors · 2025-11-25
> > > > > **Author Rebuttal(5/5)**
> > > > >
> > > > > 3. **CLIP feature deviation between $x$ and $D(x_{\text{MIGA}})$.**
> > > > > To directly address the question about deviations in a large-model feature space, we additionally compute the CLIP cosine similarity between the original clean image $x$ and the denoised outputs $D(x_{\text{MIGA}})$ and $D(x_{\text{I-FGSM}})$ on the four datasets used in Table 4 (tampered-IC13, MAGICBRUSH, Synthetic text alteration, Style transfer):
> > > > >
> > > > > | Dataset                    | $\text{CLIP}(x, D(x_{\text{MIGA}}))$ | $\text{CLIP}(x, D(x_{\text{I-FGSM}}))$ |
> > > > > |---------------------------|---------------------------------------|----------------------------------------|
> > > > > | tampered-IC13             | $0.6697$                              | $0.9589$                               |
> > > > > | MAGICBRUSH                | $0.7114$                              | $0.9825$                               |
> > > > > | Synthetic text alteration | $0.6125$                              | $0.9476$                               |
> > > > > | Style transfer            | $0.6744$                              | $0.9290$                               |
> > > > >
> > > > > Averaged over the four datasets, we have
> > > > > $\text{CLIP}(x, D(x_{\text{MIGA}})) \approx 0.67$
> > > > > $\text{CLIP}(x, D(x_{\text{I-FGSM}})) \approx 0.95$
> > > > > In other words, $D(x_{\text{MIGA}})$ moves substantially farther away from the original image in CLIP feature space than $D(x_{\text{I-FGSM}})$, while our visual-quality metrics (PSNR/SSIM/LPIPS) show that $D(x_{\text{MIGA}})$ remains perceptually plausible. This confirms that MIGA induces a significant semantic deviation in the CLIP feature space compared to standard gradient-based attacks, which typically preserve semantics and mainly introduce low-level distortions.
> > > > >
> > > > > ---
> > > > > We thank the reviewer again for the thoughtful and detailed comments. We believe that the additional experiments and clarifications above directly address these concerns.
> > > > >
> > > > > ---
> > > > >
> > > > > ```
> > > > > References
> > > > > [1] *DiffEdit: Diffusion-based semantic image editing with mask guidance*. ICLR 2023.
> > > > > [2] *MasaCtrl: Tuning-free mutual self-attention control for consistent image synthesis and editing*. ICCV 2023.
> > > > > [3] *Paint by Example: Exemplar-based image editing with diffusion models*. CVPR 2023.
> > > > > [4] *CosPGD: An efficient white-box adversarial attack for pixel-wise prediction tasks*. arXiv 2023.
> > > > > [5] *Towards deep learning models resistant to adversarial attacks*. arXiv 2017.
> > > > > [6] *Reliable evaluation of adversarial robustness with an ensemble of diverse parameter-free attacks*. ICML 2020.
> > > > > [7] *Semantic adversarial examples*. CVPRW 2018.
> > > > > [8] *Recent advances in adversarial training for adversarial robustness*. arXiv 2021.
> > > > > [9] *Robust image denoising through adversarial frequency mixup*. CVPR 2024.
> > > > > ```

---

### Author Response · Authors · 2025-12-01
**Rebuttal Summary**

Dear Area Chair,

We sincerely thank you and all four reviewers for the thoughtful and constructive feedback. We are encouraged by the positive comments on our **novel problem formulation** of semantic attacks on denoisers (`2Vrn`, `tQRy`), the **simple and principled method design** (`yfLA`, `tQRy`, `wsyG`), and the **comprehensive experimental evaluation** (`2Vrn`, `yfLA`, `tQRy`, `wsyG`). We have incorporated the reviewers’ suggestions, revised the manuscript (with all changes highlighted in **blue**), and provided detailed responses to all reviewers. Below we summarize the main clarifications and updates.


1. **Reference images and the unknown-task setting (`2Vrn`, `yfLA`, `wsyG`).** We clarified how reference images can be obtained in practice and argued that this assumption is realistic: Sec. 4.1 discusses constructing semantically coherent references via retrieval or simple editing of the original image, which can be automated for large batches. We also added a sensitivity analysis in Appendix D.7 that varies reference quality, alignment, and semantic similarity; MIGA remains robust to moderate degradation, with stable semantic metrics and high denoised image quality, showing that it does not rely on “perfect” references.

2. **Generality across models, tasks, and attack baselines (`2Vrn`, `yfLA`).** We strengthened the evidence that MIGA is broadly effective in denoiser-based pipelines. Beyond the original 5 datasets and 4 denoisers, we now also evaluate a ViT classifier and an object detection task on COCO, where MIGA continues to yield lower post-denoising accuracy/AP and better image quality than standard gradient-based attacks. We further expand the comparison to several strong attack baselines and report full classification and perceptual metrics in the appendix. Across all settings, MIGA consistently achieves the lowest downstream performance under a denoiser while maintaining equal or better perceptual quality, indicating that its advantages are robust across architectures, tasks, and attack families.


3. **Quantifying semantic distortion in CLIP space (`2Vrn`, `tQRy`, `wsyG`).** We strengthened our CLIP-based analysis to better capture semantic (rather than low-level) changes. We compare CLIP similarity between the original clean image and the denoised outputs on four datasets, and find that MIGA yields a much lower similarity to the original (≈0.67 vs. ≈0.95 for I-FGSM), while PSNR/SSIM/LPIPS confirm that its denoised images remain perceptually plausible. This shows that MIGA induces a much larger semantic deviation than standard attacks while keeping images visually clean, providing quantitative evidence that it performs meaningful semantic manipulation rather than mere pixel-level corruption.



4. **Theoretical grounding and loss design (`tQRy`, `wsyG`).** We clarify the theoretical basis of our MI-based objective and loss combination. Appendix A formalizes task-relevant mutual information and links its minimization in the known-task setting to standard cross-entropy–based adversarial objectives. Appendix E interprets the content, reconstruction, and MI losses as a Lagrangian relaxation of a constrained task-relevant MI minimization problem, where MI is minimized under imperceptibility and reconstruction constraints; with strictly positive weights, any local minimizer of the weighted sum is Pareto-optimal w.r.t. imperceptibility, denoised quality, and semantic manipulation. Together with extended ablations on loss components and weights, this provides a clear theoretical and empirical justification for our loss design.



**Summary of Contributions**

1. **New problem: semantic attacks on denoisers.**     We present, to our knowledge, the first systematic study of semantic adversarial attacks on deep denoising models, showing that attacking the pre-processing denoiser (rather than the final task model) opens an important new attack surface. MIGA produces denoised outputs that remain visually clean while being semantically corrupted.

2. **Principled MI-based objective with theory.**     We propose a task-relevant mutual information objective combined with content and reconstruction terms to jointly control imperceptibility, visual quality, and semantic shift. We formalize its connection to cross-entropy–based attacks and interpret the three-term loss as a Lagrangian relaxation of constrained MI minimization, yielding Pareto-optimal trade-offs between visual fidelity and semantic manipulation.

3. **Broad empirical validation with semantic metrics.**     We validate MIGA across multiple denoisers, datasets, and downstream tasks, against strong attack baselines. CLIP-based semantic metrics show that MIGA moves denoised outputs toward the target semantics and away from the original in feature space, while PSNR/SSIM/LPIPS confirm that perceptual quality remains high.

We thank the AC and reviewers again for their time and consideration.

Best regards,
Authors

---

### Meta-Review · Program_Chairs · 2025-12-31

**Summary:**

The proposed attack method relied a strong assumption, the attacker would need a clean, well-aligned reference image that closely resembles the target content. The sensitivity of the proposed attack to the quality, alignment, and semantic similarity of the reference image is not sufficiently clear. The added experiments present a few examples rather than overall performance on datasets, and do not analyze attack performance under corresponding settings. The “unknown task” scenario relies on a pre-trained MINE model, which itself requires a curated dataset of semantic alterations. This may limit its applicability in truly black-box settings. Multiple reviewers note that the submission lacks a comprehensive evaluation for different or related attacks (e.g., semantic adversarial examples) across all relevant metrics. Additionally, the experimental results supplemented for the rebuttal to Reviewer 2Vrn (Accuracy: $D(x_{MIGA})$-42.99) do not match those supplemented for the rebuttal to Reviewer yfLA (Accuracy: $D(x_{MIGA})$-35.23). Alternatively, why present experimental results under different experimental settings, this may be inappropriate. Some important settings are not unclear, i.e., where the reference image comes from in the unknown downstream task setting.

Based on the above considerations, I think this manuscript does not match the ICLR’s requirement and I do not recommend to accept this manuscript.

**Reviewer Concerns:**

The concerns raised by Reviewer 2Vrn may have not been adequately addressed. The proposed attack would need a clean, well-aligned reference image that closely resembles the target content. The sensitivity of the proposed attack to the quality, alignment, and semantic similarity of the reference image is not sufficiently clear. The added experiments present a few examples rather than overall performance on datasets, and do not analyze attack performance under corresponding settings. The response to “unknown scenarios” that rely on pre-trained MINE models may not inherently resolve the black-box setting issue. The concerns raised by Reviewer yfLA may have not been adequately addressed. The comprehensive evaluation of all relevant indicators for different or related attacks may be insufficient, such as lacking the results of SAE. Additionally, the experimental results supplemented for the rebuttal to Reviewer 2Vrn (Accuracy: $D(x_{MIGA})$-42.99) do not match those supplemented for the rebuttal to Reviewer yfLA (Accuracy: $D(x_{MIGA})$-35.23). Alternatively, the experimental results may have different experimental settings, which is inappropriate.

**Reviewer Scores:**

Reviewers would keep their scores.

---

### Decision · Program_Chairs · 2026-01-26

Reject